# Lateralized cerebellar connectivity differentiates auditory pathways in echolocating and non-echolocating whales

Sophie Flem[1,2]*, Gregory Berns[3], Ben Inglis[4], Dillon Niederhut[4], Eric Montie[5], Terrence Deacon[6], Karla L. Miller[7], Peter Tyack[8], Peter F. Cook[1,2,9]*

1 Psychology Department, Division of Social Sciences, New College of Florida, Sarasota, Florida, United States of America, 2 Master's in Marine Mammal Science, Sarasota, Florida, United States of America, 3 Department of Psychiatry and Behavioral Sciences, Emory University School of Medicine, Atlanta, Georgia, United States of America, 4 Henry H. Wheeler Jr. Brain Imaging Center, University of California, Berkeley, Berkeley, California, United States of America, 5 Department of Natural Sciences, University of South Carolina Beaufort, Bluffton, South Carolina, United States of America, 6 Anthropology Department, University of California, Berkeley, Berkeley, California, United States of America, 7 Wellcome Centre for Integrative Neuroimaging, FMRIB, Nuffield Department of Clinical Neurosciences, University of Oxford, Headington, Oxford, United Kingdom, 8 Biology Department, Woods Hole Oceanographic Institution, Woods Hole, Massachusetts, United States of America, 9 Institute of Marine Sciences at University of California, Santa Cruz, California, United States of America

* pcook@ncf.edu (PC); sophieflem23@ncf.edu (SF)

## Abstract

We report the first application of diffusion tractography to a mysticete, which was analyzed alongside three odontocete brains, allowing the first direct comparison of strength and laterality of auditory pathways in echolocating and non-echolocating whales. Brains were imaged post-mortem at high resolution with a specialized steady state free precession diffusion sequence optimized for dead tissue. We conducted probabilistic tractography to compare the qualitative features, tract strength, and lateralization of potential ascending and descending auditory paths in the mysticete versus odontocetes. Tracts were seeded in the inferior colliculi (IC), a nexus for ascending auditory information, and the cerebellum, a center for sensorimotor integration. Direct IC to temporal lobe pathways were found in all animals, replicating previous cetacean tractography and suggesting conservation of the primary auditory projection path in the cetacean clade. Additionally, odontocete IC-cerebellum pathways exhibited higher overall tract strength than in the mysticete, suggesting they may play a role in supporting the rapid sensorimotor integration demands of echolocation. Further, in the mysticete, contralateral right IC to left cerebellum pathways were 17x stronger than those between left IC and right cerebellum, while in odontocetes, the laterality was reversed, and left IC to right cerebellum pathways were 2-4x stronger than those between right IC and left cerebellum. This lateralization may also relate to echolocation. Right cerebellum is responsible for integrating sensory and motor signals from the left cortical hemisphere, and in odontocetes, this hemisphere

**Data availability statement:** All relevant data for this study are publicly available from the OSF repository (https://osf.io/phtje).

**Funding:** This study was financially supported by the Human Frontier Science Program (https://www.hfsp.org) in the form of a grant (RGP0019/2022) received by PFC and SF. This study was also financially supported by the Office of Naval Research (https://www.onr.navy.mil) in the form of a grant (N00014-23-1-2065) received by PT.

**Competing interests:** The authors have declared that no competing interests exist.

likely controls the contralateral right-side phonic lips, which have been empirically implicated in the production of echolocation clicks. We also found differences in the specific subregions of cerebellum targeted by the IC between the mysticete and odontocetes, some of which may also bear on hearing and vocal production. This study establishes foundational knowledge on mysticete brain connectivity and extends knowledge on pathways supporting hearing and auditory-motor integration across the order Cetacea.

## Introduction

### Cetacean brains and behavior

Cetaceans present a fascinating set of species for studying changes in brain and behavior following their evolutionary adaptation to an aquatic environment some 60 million years ago [1]. Observations of both wild and captive cetaceans suggest they possess a wide range of complex sensory and cognitive capabilities [2]. Many species engage in extensive communication with conspecifics and other cetaceans, facilitating their rich and intricate social lives, and possibly enabling the development of learned individual identity signals [3] and rudimentary culture [4,5]. The sensory, behavioral, social, and cognitive complexity observed in cetaceans has motivated scientific study of their similarly complex, large, and highly convoluted brains [6]. Though there are 94 known living cetacean species [7], most studies on neuroanatomy and behavior have involved but a small number of these species, particularly bottlenose dolphins (*Tursiops truncatus*), orcas (*Orcinus orca*), and beluga whales (*Delphinapterus leucas*) [5]. Notably, there is a particular dearth of work on baleen whales (parvorder Mysticeti). Given the evolutionary split between odontocetes and mysticetes 38.8 million years ago [8], and their divergent sensorimotor capabilities, neuroanatomical comparisons between these closely-related suborders may be particularly fruitful.

Though recent work has included important advances such as multimodal methods that integrate analyses of the gross neuroanatomical and cellular levels [9,10], the practical and ethical difficulty of carrying out non-invasive functional neuroimaging in large, fully-aquatic mammals like cetaceans has prevented researchers from clearly and conclusively elucidating the *specific functions* of the myriad regions in the great, convoluted mass of the cetacean cortex. Current understanding relies predominantly on decades-old invasive functional studies [11–14] patched together with more recent cytoarchitectural, gross anatomical, and behavioral characterization [15–20]. In other words, though the behavioral and cortical complexity of cetaceans is well-established [2], the question of *how* the complex brain functions to subserve complex behavior remains largely undetermined.

### Cetacean auditory system

The cetacean auditory system serves as an ideal starting point for understanding the mechanisms by which cetacean brains subserve their behavior. Because of the long range over which sound travels through seawater and the reduced range of

light in much of the ocean, the cetacean nervous system evolved to specialize in auditory processing [21]. There is clear evidence of rapid evolutionary adaptation of external auditory structures in the cetacean lineage [22], and the need to process complex auditory information may have contributed heavily to the dramatic encephalization seen in cetaceans [23]. Further, different suborders of cetacea have evolved to do very different things with sound; while many utilize passive audition in service of complex communication, navigation, foraging, and hunting [24], toothed whales (suborder Odontoceti) additionally evolved the capability for a more active form of audition: echolocation [25]. Echolocation, also known as sonar, occurs when an organism emits sound waves in order to gain information about features of its environment from the echoic return signals [26]. Paleontological evidence suggests that echolocation emerged in the earliest diverging odontocetes during the Oligocene, approximately 30 million years ago [27], and evolved rapidly following its initial emergence due to its affordance of plentiful access to prey across otherwise acoustically complex environments [28,29]. Across mammalian species, subcortical auditory integration is time-pressured and requires very rapid processing [30], time demands that are likely compounded by the high frequencies of echolocation signals in dolphins. New capacities to hunt in muddy rivers, shallow icy waters, and high-pressure ocean depths likely provided the caloric payoffs requisite to carry out this energetically-demanding, time-pressured processing [31–32]. The continuous refinement of echolocation capacities in odontocetes has persisted to the present day, as can be seen in neuroanatomical studies reporting the immense hypertrophy of the modern-day odontocete subcortical auditory system [19,21,33]. The absolute size of the dolphin medial geniculate nucleus (MGN) is seven times larger, the inferior colliculi (IC) twelve times larger, and the lateral lemniscus *250* times larger than in the comparably-sized human brain [34,35]. The bulk of prior research on auditory adaptations in cetaceans has focused on the diencephalon, mesencephalon, or myelencephalon, where brain structures and their connectivity patterns are more conserved across clades. However, as discussed above, difficulties in collecting functional data on the cetacean neocortex has rendered the task of identifying the cortical components of the auditory system quite difficult.

Foundational studies establishing the locations of sensory projection fields in cetacean neocortex employed invasive electrophysiological methods that directly measured which regions of cortex responded to different types of sensory input. These studies initially localized the primary auditory projection field (A1) in a long "belt" of dorsal cortex in the suprasylvian gyrus, near the vertex of the brain, and directly lateral to the primary visual cortex [14,36,37]. Despite one prior study with opportunistic evidence of electrophysiological response to sound in the temporal lobe [38] and subsequent tracing identifying medial geniculate projections to the temporal operculum [39], later studies tended to accept the localization of a mammalian suprasylvian A1 [18,35,40]. This convention was challenged in Berns et al. [41], a first-of-its-kind study that assessed auditory connectivity in opportunistically-collected post-mortem dolphin brains. Berns et al. [42] used a diffusion-weighted steady state free precession (DW-SSFP) protocol specialized for post-mortem imaging [43] and conducted diffusion tensor imaging (DTI) tractography on the resulting diffusion-weighted images to allow analysis and visualization of cerebral white matter tracts [44]. Critically, diffusion tractography allows some inference of functional connectivity from structural connectivity, thus granting researchers a window into cetacean brain function, albeit an indirect one [45–47]. Berns et al. [41] reported that in the dolphins studied, the ascending auditory pathway projected not to the suprasylvian gyrus, but instead to a region in the deep temporal lobe, i.e., the more typical location of mammalian A1. This suggested that there may be multiple topographically distinct cortical centers for auditory processing in the dolphin. In this case, the dorsal electrode insertions used in the original electrophysiology studies on cortical auditory processing may have simply missed this site due to its harder-to-reach location in deep temporal lobe. Thus, it is possible that these earlier invasive studies were in fact measuring suprasylvian auditory processing signals that were "downstream" of temporal A1. Indeed, in Berns et al. [41] seeding the inferior colliculi produced bilateral paths to deep temporal lobe, and in turn, seeding these IC-targeted temporal lobe regions produced bilateral paths to previously-identified suprasylvian auditory centers. Odontocete auditory processing may thus parallel the case in bats, in which temporal A1 sends auditory information to postero-dorsal centers that process finer temporal and spatial features of sound [48]. In the mysticetes, meanwhile, neither electrophysiological nor tractographical methods have yet been applied to study ascending auditory pathways.

## Lateralization

Following Berns et al. [41], Wright et al. [42] used diffusion-weighted MRI, though with a standard echoplanar protocol, to examine the lateralization observed in the major white matter tracts in the brain of a *Tursiops truncatus* specimen. Wright et al. [42] found evidence for substantial cortical asymmetry, particularly in putative auditory and vocal tracts. This finding is in line with prior evidence of sparse interhemispheric connectivity in cetaceans and, relatedly, unihemispheric sleep [49–51]. In addition, in at least some species of echolocating odontocetes, including beluga whales and dolphins, individuals have two sets of lateralized phonic lips that are employed differentially depending on function: while the left lips are responsible for social communication, the right lips are responsible for high-frequency echolocative clicks [52,53]. Importantly, Ames et al. [52] report that *function*, not *frequency*, determines which phonic lips are employed to produce sound, as even high-frequency social buzzes are typically performed by the left lips, in addition to low-frequency social whistles.

The lateralization of phonic lip function, also known as the functional laterality hypothesis, provides a new foundation upon which to bridge cetacean behavior with brain function. Despite limited direct data on functional lateralization in cetacean cortex, the principle of contralateral organization, in which each half of the forebrain and thalamus has efferent and afferent connections to the opposite side of the body [54], applies broadly to vertebrate species all the way back to the Ordovician period [55]. It is safe to presume that cetacean nervous systems are organized contralaterally as well. In many species, vocal motor output is bilateral and under control of non-volitional subcortical regions and nuclei (e.g., periaqueductal gray and nucleus ambiguus) [56]. In species with volitional control of vocal motor behavior such as humans, there is direct cortical contribution to vocal motor output [57]. There is robust behavioral evidence that dolphins can flexibly control their vocal motor output [58], suggesting at least some aspects of output are volitional and cortically-controlled. Taken together, this suggests that via application of the principle of contralateral cortical control, the anatomical and likely functional left-right division of the odontocete vocal motor system may offer a rare window into cetacean neuroanatomy, as well as insight into the neurobiological underpinnings of a highly unique form of sound production. If, in dolphins and beluga whales, the left phonic lips are employed for social functions and the right phonic lips for echolocation, then the contralateral cortical hemispheres may be accordingly specialized to process social and echolocative information, too– and such functional neural lateralization is particularly plausible given the previous evidence of hemispheric independence in cetaceans. Such a hypothesis may be investigated by studying efferent motor pathways from the central nervous system to the phonic lips, and by studying the pathways prior to and parallel to these that prepare and refine the appropriate motor information via the integration of auditory, premotor, and social or navigational information. In other words, investigation of this hypothesis may require analysis of not only ascending auditory pathways, but also *descending* auditory pathways, the latter of which have been relatively unexplored in cetaceans thus far.

## Descending "acousticomotor" pathways

Although the classical representation of the auditory system follows an ascending pathway from brainstem nuclei to inferior colliculi and then medial geniculate nucleus before reaching A1, recent data from human and animal models shows strong evidence of dense reciprocal connections, ascending and descending between every major waystation on the auditory pathway [59–62]. The inferior colliculus is central in this reciprocal architecture, serving as the termination point for fibers ascending from the brainstem auditory nuclei while also receiving heavy top-down innervation from primary auditory cortex [60]. The top-down connectivity allows for feedback from auditory cortex onto the IC and to the brainstem auditory nuclei, with potentially faster transit time from cortex to brainstem than vice versa [59]. This suggests that top-down cortex to IC pathways could play an essential role in modulating early auditory processing. In addition, descending acoustic information from the cortex and IC has been shown to densely target the cerebellum [63,64].

 

In general, the cerebellum is believed to integrate sensory information with motor plans to compute a feedforward prediction of the best solution to any particular sensorimotor situation, including volitional motor output [65]. While descending motor signals from primary motor cortex pass directly through the pyramids to the spine or, in the case of facial motor processing, relevant brainstem nuclei, the cerebellum receives efference copy from primary motor cortex and projects back to upper motor neurons in the cortex and subcortex to refine and coordinate complex motor output [66]. Thus, the cerebellum's afferents and efferents constitute a parallel motor plan refinement pathway for volitional motor output, including vocal motor output in humans [67,68]. Because of the role of the cerebellum in using cortical sensory afferent information to determine how to refine motor output, Huffman & Henson [59] dubbed auditory-to-cerebellar pathways "acousticomotor." Here, we will refer to them as descending auditory pathways or IC to cerebellum connections, because their exact role in motor refinement is still unknown. Importantly, along with sensorimotor processing, evidence suggests the cerebellum has also been found to play a role in general audition [69] and speech processing in humans [68,70,71]. A picture is now emerging of the role the cerebellum plays in making multimodal feedforward predictions that are not restricted to motor processing [65,72].

The cerebellum is of particular interest in cetacean neurobiology due to its clear hypertrophy, particularly among delphinids [6]. In common (*Delphinus delphis*), bottlenose, and Atlantic white-sided dolphins (*Lagenorhynchus acutus*), the cerebellum makes up an average of approximately 15% of total brain volume, as compared to the 10% and 9% of total brain volume observed in humans and non-human primates, respectively [23,73,74]. While generally understudied, one explanation for cerebellar hypertrophy in dolphins is that it is involved in some way in vocal and echolocation processing [75]. This hypothesis would parallel findings in bats, which also have relatively large cerebella compared to evolutionarily related non-echolocating rodents [76]. Electrophysiological recording in bats has identified cells in bat cerebellum that play a specific role in echolocation, helping predict target location [77]. The role of descending auditory tracts in dolphins requires further exploration, and particularly with respect to the expanded cerebellum, given its potential role in auditory processing and echolocation. Furthermore, dolphins can apparently remain continuously vigilant for weeks at a time by using echolocation during unihemispheric sleep [78], suggesting that sufficient echolocation processing is possible in either hemisphere (despite functional laterality at the level of the phonic lips), *or* that bilateral or contralateral subcortical and cerebellar mechanisms may support the capability for echolocation during unihemispheric cortical sleep [79].

In sum, while DTI tractography has been used to investigate ascending auditory pathways [9,41] and lateralization [42] in *toothed whales*, to date, it has not been employed in the analysis of descending auditory pathways that may be critical for echolocation, and it has never been employed to study baleen whale brains. Baleen whales have a partially conserved laryngeal sound production mechanism [80], which is quite different from the nasal sound production mechanisms of the odontocetes [81]. These evolutionary differences in form and function may be linked to differences in descending auditory pathways in particular. Application of DTI tractography to mysticetes could provide foundational knowledge on ascending *and* descending auditory pathways of this elusive, understudied parvorder. Furthermore, study of the mysticete auditory system may offer clues as to what features of the odontocete auditory system are specialized for echolocation, as the capability of odontocetes for echolocation is one of the most salient features that differentiates them from the closely related mysticetes.

In the present study, the successful acquisition and diffusion-weighted MRI imaging of an intact sei whale brain (*Balaenoptera borealis*) enabled the first application of DTI tractography to a mysticete. The *B. borealis* specimen was analyzed alongside brains opportunistically collected from odontocetes, including a common dolphin, pantropical spotted dolphin (*Stenella attenuata*), and Atlantic white-side dolphin. In order to establish foundational knowledge on the structural connectivity of mysticete brains, as well as search for differences from odontocetes that may shed light on the neural basis of the closely-related parvorder's echolocative capabilities, this study employed probabilistic tractography to compare the qualitative features, tract strength, and lateralization of potential ascending (IC to temporal lobes) and descending (IC to cerebellum) auditory paths in mysticetes and odontocetes.

## Results

### Whole brain metrics: Volume and fractional anisotropy

The whole brain volume of *D. delphis* was 1,141.51 cm³, while *S. attenuata* was 933.20 cm³, *L. acutus* was 1067.08 cm³, and *B. borealis* was 2,864.37 cm³ (Table 1). The inferior colliculi made up 0.28% of the total brain volume in *D. delphis*, 0.37% in *S. attenuata*, 0.34% in *L. acutus*, and 0.28% in *B. borealis* (Table 1). The cerebella made up 17.47% of the total brain volume in *D. delphis*, 16.54% in *S. attenuata*, 12.74% in *L. acutus*, and 13.21% in *B. borealis* (Table 1). There were no grossly obvious differences between the relative volumes of the right and left inferior colliculi or cerebella in any of the animals studied. In *D. delphis*, the mean fractional anisotropy (FA) value was 0.093; in *S. attenuata*, mean FA was 0.100; in *L. acutus,* 0.123; and in *B. borealis,* 0.170 (Table 1).

### Tract strength

Each tractography output, abbreviated in this paper as "trace," included a *waytotal* value denoting the number of stream-lines that satisfied the tractography algorithm by originating in the seed region of interest (ROI) and met waypoint and/or exclusion criteria (Thaploo et al., 2022; see Fig 1a for a visual overview of the ROIs and waypoint/exclusion criteria for each trace). We then calculated the absolute tract *strength* of each trace by dividing its waytotal by the total brain volume of the respective specimen (Fig 1b).

Among all specimens, the greatest within-specimen tract strength was observed in IC to ipsilateral cortex traces, with the exception of *D. delphis*, in which the left IC-right cerebellum trace showed the strongest within-specimen tract strength. Among all four specimens, the greatest *bilateral* tract strength, and the greatest tract strength overall, was observed in the IC to ipsilateral cortex traces of *B. borealis*, where the volume-adjusted tract count for left and right side traces totaled 1546.15 and 1352.53, respectively. Between all specimens, the weakest observed tract was between the left IC and right cerebellum in *B. borealis*. After dividing by whole-brain volume, we found a mere 5.21 streamlines in this trace for *B. borealis*, as opposed to the 811.31 calculated for the same trace in *D. delphis*. With the exception of the left IC to right cerebellum trace in *B. borealis*, the weakest within-specimen tract strength was found in contralateral IC to cortex traces. The weakness of contralateral IC to cortex traces was particularly pronounced in the *L. acutus* specimen, though notably, the tract strength observed in this specimen generally tended to be lower than the other specimens.

### Quantitative lateralization

The *laterality* of each trace was quantified by two methods: simple division of the larger waytotal by the smaller, and calculation of the *laterality index* for each (Wright et al., 2018; Vernooij et al., 2007) [42,82] (Table 2, Fig 1b). In the following, "right-lateralized" versus "left-lateralized" designations reflect which cortical or cerebellar hemisphere was *targeted* more strongly by a respective inferior colliculus, rather than designating whether right or left IC were seeded.

**Table 1. Mean FA and volumetrics.** Mean fractional anisotropy (FA), whole brain volume (WBV), percentage of whole brain volume contained in inferior colliculi (IC/ WBV) and percentage of whole brain volume contained in cerebella (Cerebella/ WBV) for each specimen.

|  | *D. delphis* | *S. attenuata* | *L. acutus* | *B. borealis* |
|---|---|---|---|---|
| *Mean FA* | 0.093 | 0.100 | 0.123 | 0.170 |
| *WBV* | 1,141.51 | 933.20 | 1,067.08 | 2,864.37 |
| *IC/ WBV* | 0.28% | 0.37% | 0.34% | 0.28% |
| *Cerebella/ WBV* | 17.47% | 16.54% | 12.74% | 13.21% |

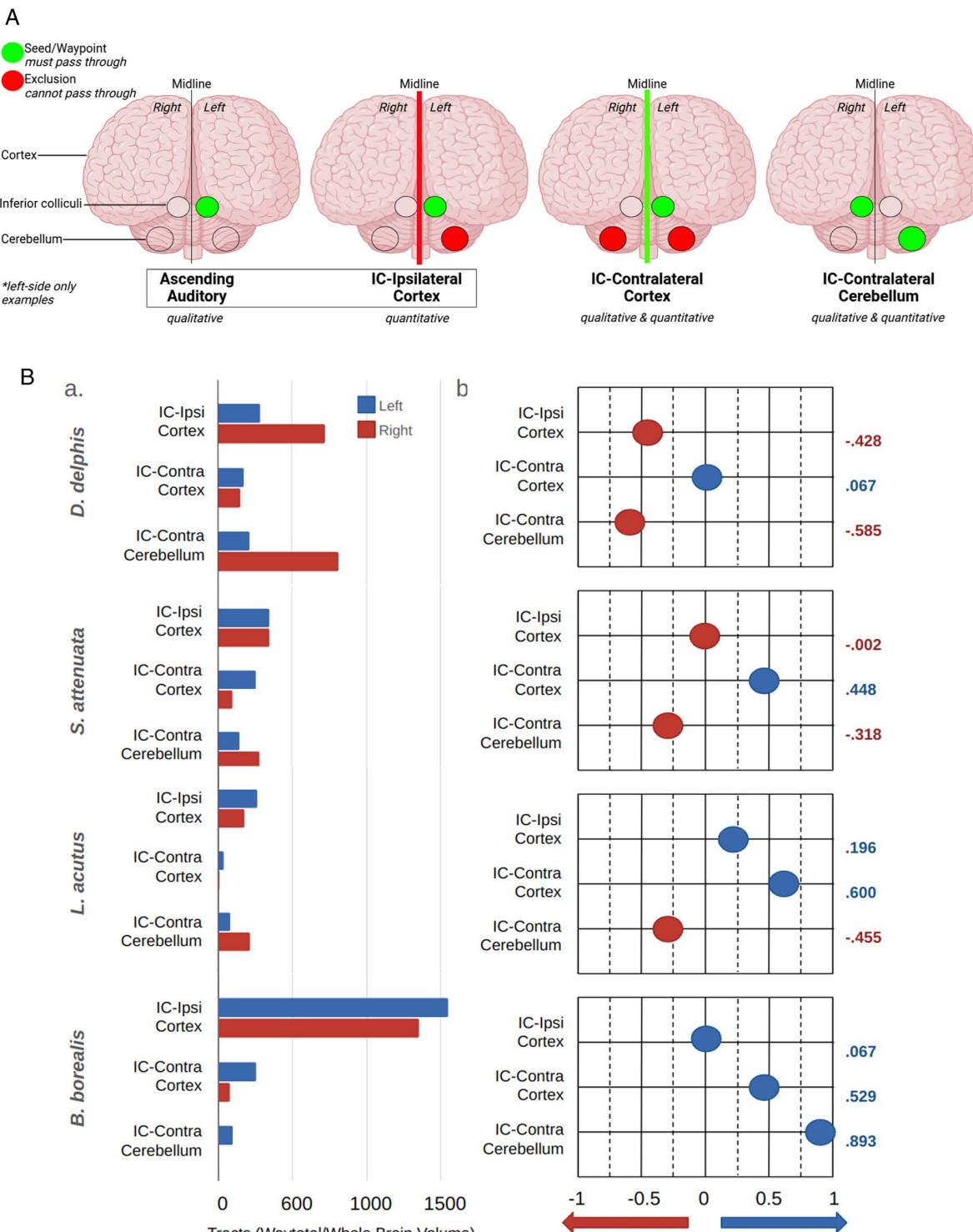

**Fig 1. a. Overview of ROIs and waypoint/exclusion criteria for each trace.** In all traces, the inferior colliculi (IC) were the primary seed/ROI through which pathways had to pass. Ascending auditory traces simply show the results of a plain IC-seed protocol, and are typically shown/discussed for their qualitative information they communicate about the brain regions that connect to IC. IC-ipsilateral cortex is the corresponding trace used for *quantification* of how many IC pathways project to the ipsilateral cortex on each side; any pathways that traverse cerebellum or cross the midline were excluded. IC-contralateral cortex traces included only pathways that crossed the seeded IC and the midline but did *not* traverse the cerebellum, and are presented

in both quantitative and qualitative results. IC-contralateral cerebellum traces included pathways that traversed IC and the cerebellum contralateral to the seeded IC, and are also presented in both qualitative and quantitative results. Green signifies that a region was used as a seed or waypoint, i.e., pathways *had* to contact them in order to be included, while red signifies that a region was excluded, i.e., pathways that contacted them were excluded. Only left-side traces are shown in this figure, but each of these protocols were performed for both right and left sides. **b. Tract strength and laterality index.** Panel (a) contains bar graph representations of the corrected tract strength of each trace in each specimen. See Table 2 for exact values. Panel (b) contains graphical representations of the laterality index (LI) of each trace in each specimen (LI formula and LI graph design adapted from Wright et al. [42]). LI values range from -1, maximally right-lateralized, to 1, maximally left-lateralized, and are given to the right of each graph. Refer to Fig 1a for details on the ROIs and criteria for each trace. IC = inferior colliculi.

In simple traces of IC connectivity, no lateralization effects were observed (S8 Tables in S1 File), in accordance with the lack of volumetric differences found between left and right IC in any of the specimens (Table 1). Followup traces examining connectivity between each IC and respective ipsilateral cortex– i.e., excluding both cerebella and the respective cortical hemispheres contralateral to each IC– yielded only mild lateralization effects in all specimens other than *D. delphis* (Table 2). In *D. delphis*, a more notable lateralization effect was observed, with 2.50 times more connections found between the right IC and its ipsilateral cortex than the left (Table 2). In traces between each IC and its respective *contralateral* cortex– i.e., with both cerebella excluded and the sagittal-plane midpoint of each brain set as a waypoint–a common pattern of quantitative lateralization was found in all specimens, echolocator or not (Table 2). Comparatively robust contralateral cortical connectivity of the right IC was observed in all specimens, most strongly in *L. acutus* and *B. borealis*, more moderately in *S. attenuata*, and most weakly in *D. delphis* (Table 2). In *L. acutus*, the contralateral connectivity of the right IC to left cortex was 4.00 times greater than the left IC's connectivity to right cortex (Table 2).

The strongest lateralization effects were found in traces between inferior colliculi (IC) and the respective cerebellar hemispheres contralateral to each (Table 2). In these traces, we observed a lateralization pattern that was consistent between odontocetes yet robustly reversed in the mysticete (Table 2). Indeed, the most strongly lateralized value in this dataset comes from the *B. borealis* specimen that displayed connectivity between its right IC and left cerebellum that was 17.77 times stronger than between its left IC and right cerebellum (Table 2). By contrast, all odontocetes exhibited greater connectivity between their left IC and right cerebellum (Table 2). In *D. delphis*, this right-lateralization effect was most pronounced, with 3.82 times more connections found between left IC and right cerebellum than right IC and left cerebellum (Table 2). Meanwhile, in *L. acutus* and *S. attenuata*, left IC-right cerebellar tracts were 2.67 and 1.93 times stronger, respectively, than their contralateral counterparts (Table 2).

**Table 2. Tract strength and laterality factor.** Tract strength (i.e., waytotal divided by whole brain volume) and lateralization factor (i.e., how many times more tracts were observed in one hemisphere versus the other) for each trace in each specimen. Ipsi = ipsilateral, Contra = contralateral, IC = inferior colliculi, and LF = lateralization factor. Right versus left designations denote which cortical or cerebellar hemisphere was targeted more strongly by a respective inferior colliculus, rather than which the side of the IC was seeded. Refer to Fig 1a for details on the ROIs and criteria for each trace.

| | *D. delphis* | | | *S. attenuata* | | | *L. acutus* | | | *B. borealis* | | |
|---|---|---|---|---|---|---|---|---|---|---|---|---|
| | Left | Right | LF | Left | Right | LF | Left | Right | LF | Left | Right | LF |
| **IC - Ipsi. Cortex** | 287.911 | 719.000 | **2.497x Right** | 349.273 | 350.586 | 1.004x Right | 262.506 | 176.421 | 1.488x Left | 1546.153 | 1352.539 | 1.143x Left |
| **IC - Contra. Cortex** | 175.296 | 153.161 | 1.145x Left | 257.329 | 98.088 | **2.623x Left** | 37.535 | 9.374 | **4.004x Left** | 253.613 | 78.065 | **3.249x Left** |
| **IC - Contra. Cerebellum** | 212.354 | 811.316 | **3.820x Right** | 145.350 | 281.023 | **1.933x Right** | 79.636 | 212.374 | **2.667x Right** | 92.614 | 5.213 | **17.767x Left** |

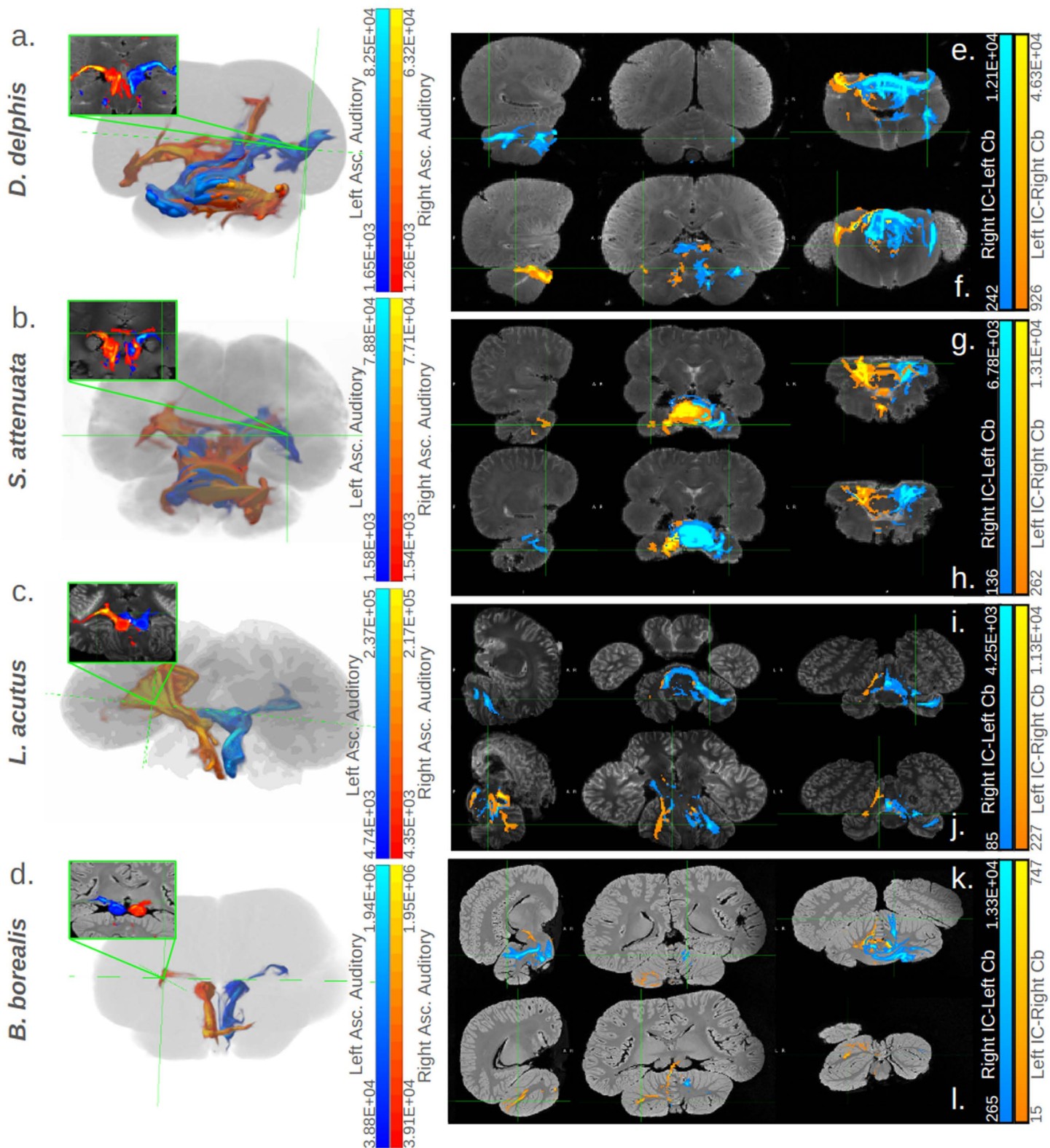

**Fig 2. Cortical and cerebellar targets of inferior colliculi.** Panels (a)-(d) contain 3D representations of ascending auditory tracts for each specimen. Tracts seeded in the right inferior colliculi (IC) are colored red, while those seeded in the left are colored blue. Adjacent to the 3D view, bright green

boxes display coronal cross-section views of these traces in each specimen, highlighting their path through the ventral thalamus (putative medial geniculate nucleus; for a more detailed view see Fig 3.) Panels (e)-(l) contain orthographic views of some primary cerebellar targets of traces between the IC and cerebella in each specimen. Tracts targeting the right cerebellum are colored orange, while those targeting the left are colored turquoise. For each specimen, two orthographic-view images are given: one highlighting a prominent projection site in the left cerebellum, the other highlighting a prominent projection site in the right. IC = inferior colliculi, Cb = cerebellum. For more details on cerebellar projection sites, see Fig 6, S9 Table, S1 Text, S6 Figures or S7 Figures in S1 File. In all panels, the correspondence between the number of predicted tracts and the colors displayed is given in the gradient color bar on the right edge, with brighter colors indicating a higher number of tracts predicted in a given voxel and vice versa.

## Qualitative features

**Ascending auditory tracts.** In all specimens, we found bilateral pathways from inferior colliculi (IC) to temporal lobe that passed through a ventral-caudal portion of the thalamus, likely the medial geniculate nucleus (Figs 2a–d, Fig 3).

In *B. borealis*, left and right IC tracts proceeded from the deep temporal lobe adjacent to the IC to an equally ventral but more caudal-medial region of cortex in a fairly linear projection (Fig 3b). In all odontocete IC traces, this deep temporal lobe to ventro-medial-caudal cortical projection was mirrored to at least some extent, most robustly so in *S. attenuata* (Fig 4d). However, the odontocetes also showed multiple other cortical projection sites in addition to the sole ventro-caudal one observed in the mysticete (Fig 5). Consistently, all odontocetes showed bilateral projections from the IC-adjacent deep temporal lobe to a more rostral though equally ventral cortical region (Fig 4b). In *D. delphis* in particular, these rostro-ventral projections were robust, particularly on the left side (Fig 4b). These rostral-ventral tracts appeared to reach the caudate head in all odontocetes, and while the left side of the caudate was more strongly implicated across all three, in *S. attenuata*, the contralateral right IC appeared to target the left caudate more prominently than the left IC (Fig 4b). In *L. acutus*, at a fork between the rostral-bound and caudal-bound projections from the deep temporal lobe, a third type of projection ran perpendicular to these toward a more lateral region of cortex, most robustly in the right hemisphere (Fig 4a). In *D. delphis*, these lateral projections were also seen, though more prominently in the left hemisphere (Fig 4a). Additionally, a dorsal-rostral region in the left cerebral cortex of *D. delphis* exhibited uniquely robust projections from both left and right IC, though the projections from the contralateral right IC were particularly pronounced (Fig 4c).

**Contralateral collicular-cerebellar tracts.** We identified cerebellar projection sites for the IC to cerebellum traces in each specimen, using the cerebellar divisions and nomenclature established in Schmahmann et al. [95] and subsequently applied to a dolphin brain in Hanson et al. [83] as anatomical guidelines. A summary of the findings is provided below in-text, as well as in Fig 6 and S9 Table in S1 File, in which the putative functions of each projection site are further specified. For more detailed descriptions of the subcortical and cerebellar targets of the IC-cerebellar traces, refer to S1 Text in S1 File.

In all specimens, IC to cerebellar traces tended to produce bilaterally robust pathways through the auditory brainstem regions between and proximal to the pons and cerebellum, including the olivary complex, the trapezoid body, and the lateral lemniscus (Figs 2e-l, Fig 7). In all of the dolphins, multiple distinct, seemingly transverse pathways emerged rostrally from the more caudal brainstem at different levels of the pons, typically one at a dorsal extreme of the pons, one at a ventral extreme of the pons, and a third medial to these two in the dorsal-ventral axis (Fig 7, S1 Text in S1 File). In *S. attenuata*, these three parallel transverse pontine pathways, as well as the dense brainstem pathways caudal to them, tended to be fairly bilateral (Fig 7b, S6 Figures B1-4 in S1 File). Indeed, more so than in any other specimen, both IC to cerebellar traces in *S. attenuata* displayed prominent dorsal-ventral projections through the mylencephalon, possibly traversing the inferior cerebellar peduncle, that proceeded ventrally and laterally from the floor of ventricle IV on the caudal brainstem, potentially contacting the facial colliculi situated here, to the caudal border of the inferior olives and eventually the rostral-most portion of the spinal cord (Fig 7b). Interestingly, though, the left IC to right cerebellum trace, but *not* the right IC to left cerebellum trace, produced a second, similarly continuous and robust projection between the floor of ventricle IV and the caudal edge of the inferior olives, medial and slightly rostral to the bilateral parallel projections described above, closer to the facial motor nucleus (Fig 7b).

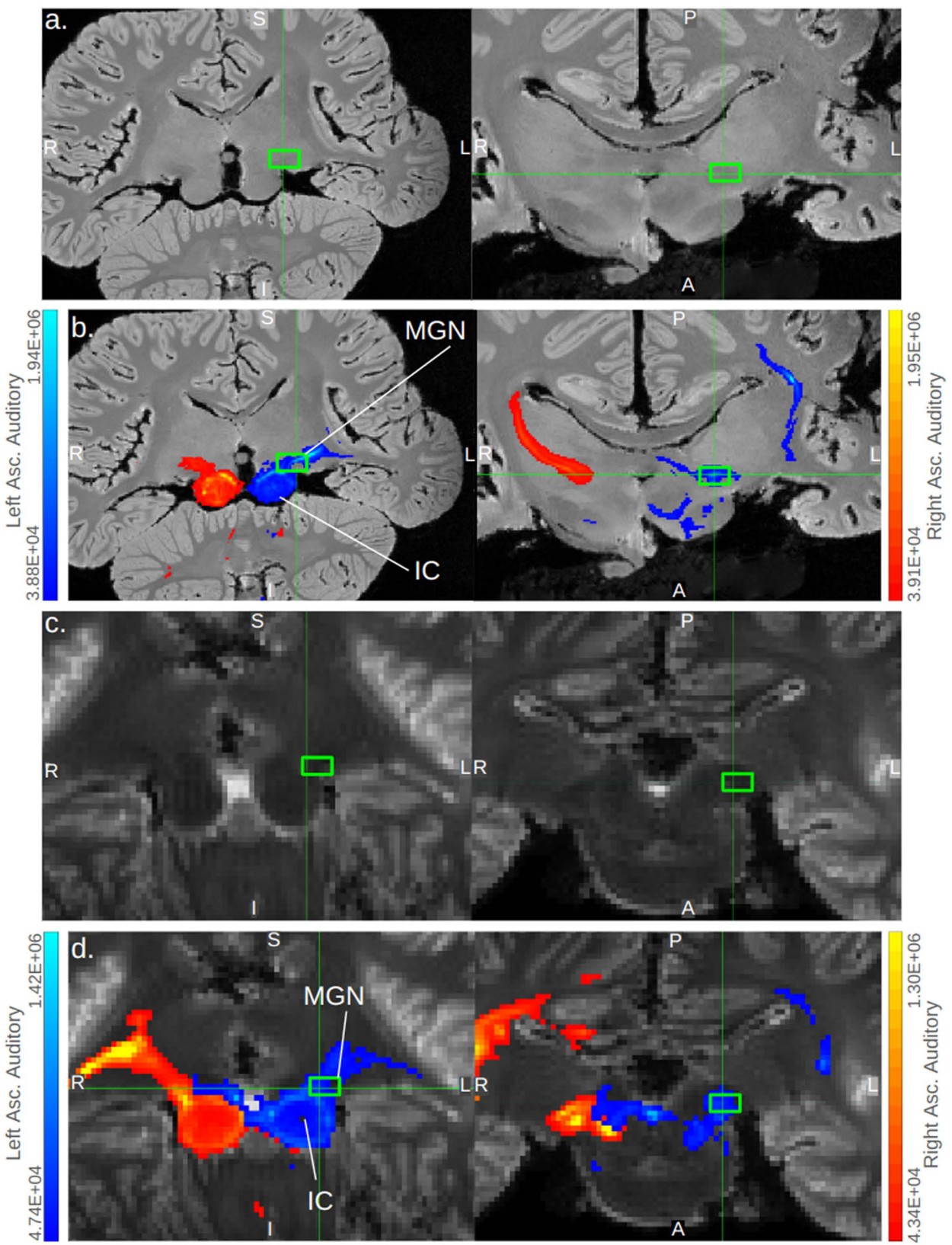

**Fig 3. Ascending auditory pathways transit putative MGN.** Coronal and axial cross-sections depicting the ventral thalamic transit of ascending auditory pathway traces in *B. borealis* (a, b) and *L. acutus* (c, d). Cross section images without (a, c) and with (b, d) tracts are shown; for *B. borealis* (a, b), a high-resolution T1-weighted image is shown, while a low-resolution b0 image is shown for *L. acutus* (c, d). Tracts are thresholded with minimum and maximum values of 0.1% and 5% of the trace waytotal, respectively, in *B. borealis* (b) and with minimum and maximum values of 1% and 30% of the trace waytotal, respectively, in *L. acutus* (d). The correspondence between predicted number of tracts in a given voxel and displayed colors of tracts is given in the gradient color bar on the right edge of panels b and d. Bright green boxes mark putative medial geniculate nucleus in each panel. MGN = medial geniculate nucleus and IC = inferior colliculi.

In the other dolphins, such instances of qualitative lateralization were often apparent. Strikingly, in the left IC to right cerebellar trace of *D. delphis*, the most dorsal of the pontine pathways continuously connected a dorsal-rostral portion of the right cerebellum, likely in lobule IX, with a dorsal-rostral extreme of the left cortex, passing through rostral extremes of diencephalic and mesencephalic structures such as the crus cerebri and substantia nigra in between (Fig 7a, S6 Figures A1-4 in S1 File). Meanwhile, in *L. acutus*, right IC to left cerebellum traces tended to produce more bilateral tracts caudal to the pons in the auditory centers of the brainstem, appearing on both right and left sides of the brainstem and cerebellar peduncles, while the left IC to right cerebellar traces remained more concentrated on the right side (Fig 7c). Compared to the dolphins, in *B. borealis*, these pontine pathways tended to be slightly more ventrally-shifted, and were more robust in right IC to left cerebellar traces as opposed to left IC to right cerebellar ones (Fig 2k-l, S6 Figure D1, S7 Figure D2 in S1 File).

In the cerebellum itself, a handful of subregions emerged as the most relevant targets of IC-cerebellar traces among all specimens. The most prominent regions were Crus I and lobule IX, which received projections from IC-cerebellar traces in all four specimens (S9 Table in S1 File). Crus I showed consistent patterns of lateralization, with left Crus I implicated in right IC to left cerebellar traces in both the mysticete and odontocetes (S1 Text, S9 Table in S1 File, Fig 6, S7 Figures in S1 File). Lobule IX was bilaterally implicated in the mysticete, while in the dolphins, the right IC to left cerebellum traces more prominently targeted left lobule IX (S1 Text, S9 Table in S1 File, Fig 6, S7 Figures in S1 File). The next most prominent subregions were lobule VIIIa and Crus II, with each receiving projections from three of the four specimens (S9 Table in S1 File). Crus II was only targeted in the odontocetes and *not* in the Mysticete, and in all odontocetes, this was observed *only* in the right IC to left cerebellar traces, i.e., in left Crus II (S1 Text, S9 Table in S1 File, Fig 6, S7 Figures in S1 File). Lobule VIIIa was targeted in all specimens except *S. attenuata*, and was bilaterally implicated in *B. borealis* and *L. acutus*, while *D. delphis* exhibited right lobule VIIIa projections only (S1 Text, S9 Table in S1 File). Finally, lobules VI, VIIb, VIIIb, and the vermis were all observed as prominent targets in two of the four specimens (S1 Text, S9 Table in S1 File). Lobule VI was implicated in the left cerebellum of the mysticete, but in the right cerebellum of one odontocete (S1 Text, S9 Table in S1 File). Meanwhile, lobule VIIb received bilateral targeting in the mysticete, but only received projections in the right cerebellum of its odontocete representative, *D. delphis* (S1 Text, S9 Table in S1 File). By contrast, lobule VIIIb was only targeted in odontocete specimens, and in both of these was bilaterally implicated (S1 Text, S9 Table in S1 File). Likewise, the vermis only received projections from IC to cerebellar traces in odontocete specimens, though in *L. acutus* the vermis was targeted more by right IC to left cerebellar traces as opposed to bilaterally (Fig 7c, S1 Text, S7 Figures C1-2 in S1 File in S1 File). Notably, anterior lobules I-V were not meaningfully targeted by IC-cerebellar traces in any of the specimens, and lobule X was prominently and bilaterally targeted, with both cerebella receiving projections from both IC, in *D. delphis*, but was not pronouncedly targeted in any other specimens (S1 Text, S7 Figure A1 in S1 File).

In all odontocete specimens, traces between IC and contralateral cerebella included some cortical projections at the more and less stringent thresholds specified above (S6 Figures A1-4, B1-4, and C1-4 in S1 File). In the mysticete, cortical projections were sparse at the liberal threshold and nearly absent at the more conservative one (S6 Figures D1-4 in S1 File). Among the odontocetes, cortical projections appeared weakest in *S. attenuata* and strongest in *D. delphis* and *L. acutus*, and were qualitatively lateralized in all, but most strikingly so in the latter specimens, with more notable projections

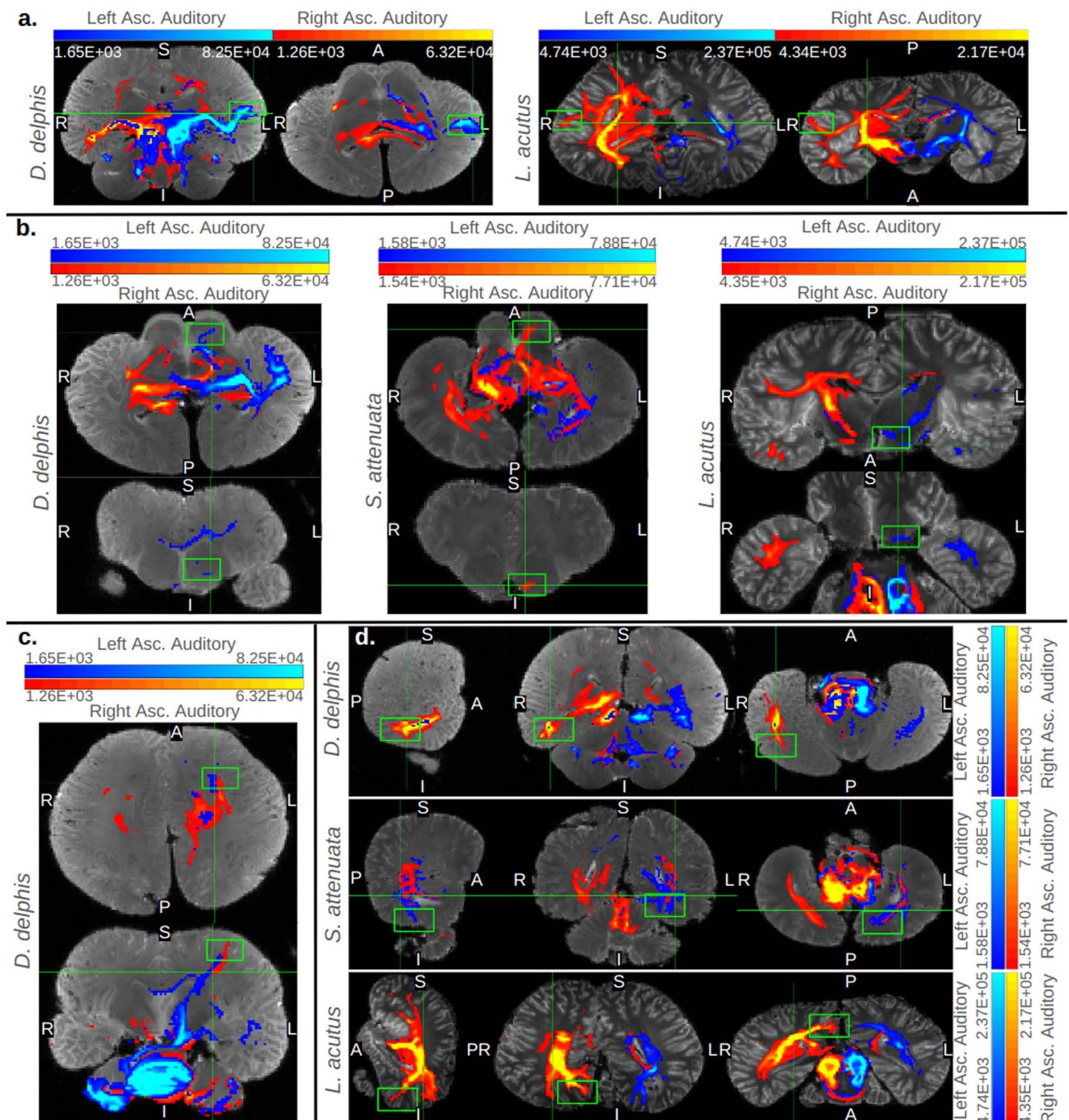

**Fig 4. Cortical targets of ascending auditory traces in odontocetes.** Orthographic cross-sections displaying key cortical targets of the inferior colliculi in the odontocete specimens. Each trace is displayed with minimum and maximum values set to 0.1% and 5% of its waytotal, respectively. In all

panels, traces from right IC are colored red and those from left IC are colored blue. Gradient color bars specify the correspondence between predicted number of tracts in a given voxel and displayed colors of tracts. Panel (a) displays coronal and axial views of projections to lateral extremes of the left and right cortex in *D. delphis* and *L. acutus*, respectively. Panel (b) displays axial and coronal views of tracts transiting the basal ganglia, potentially caudate nucleus, in all odontocetes. Panel (c) displays axial and coronal views of continuous projections between both colliculi and a dorsal-rostral region of left cortex in *D. delphis*. Finally, panel (d) displays sagittal, coronal, and axial views of projections to a caudal, ventral, and medial cortical area in all odontocetes. Bright green boxes mark the notable cortical projection sites mentioned in-text.

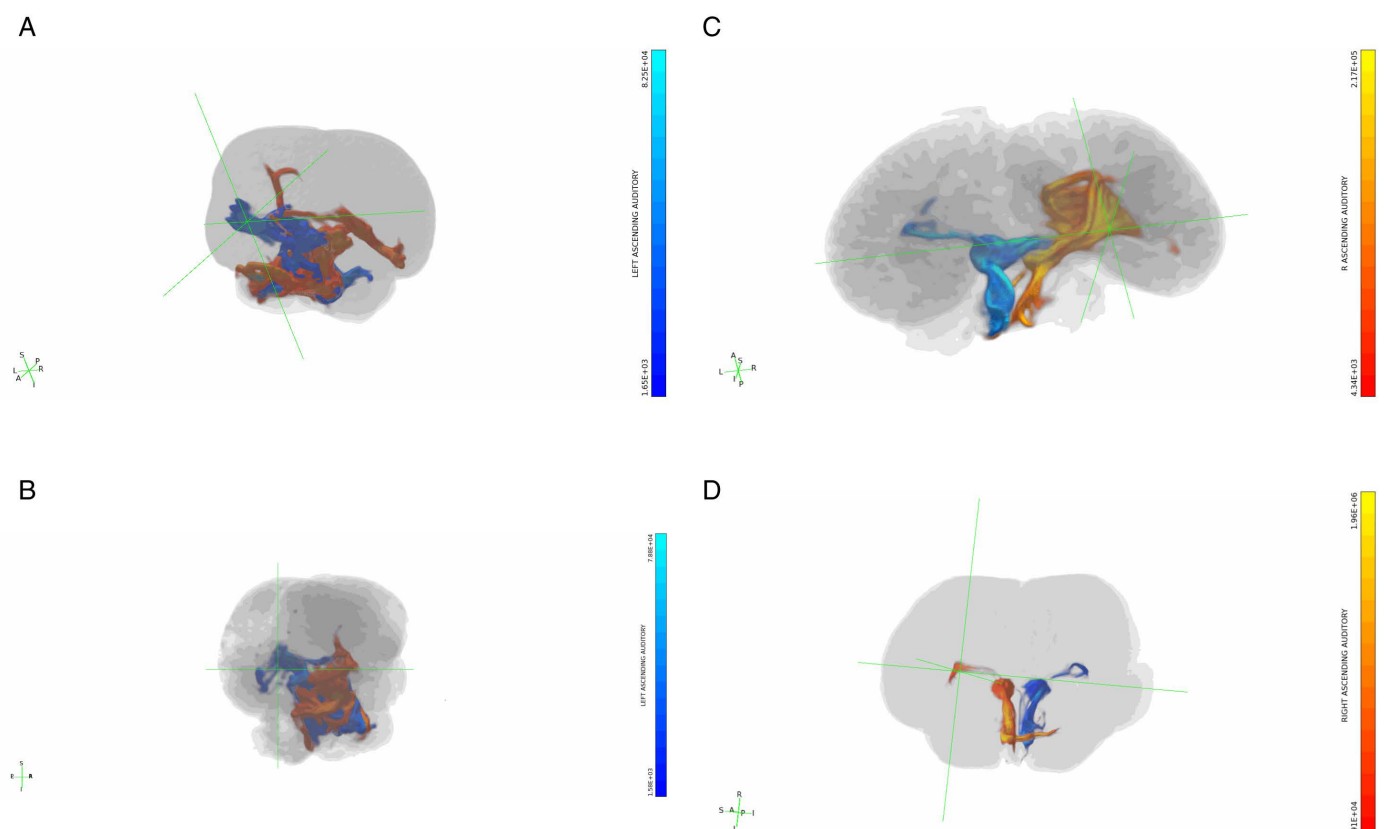

**Fig 5. Rotating 3-dimensional tractograms of ascending auditory pathways in all specimens. a.** *D. delphis*. **b.** *S. attenuata.* **c.** *L. acutus*. **d.** *B. borealis*. **a-d.** All displayed tractograms were seeded in the IC (refer to Fig 9 or S3 Figures in S1 File to view the masks of IC that served as seeds). These traces are displayed with minimum and maximum values set to 0.1% and 5% of their waytotals, respectively. In all panels, traces from right IC are colored red and those from left IC are colored blue. Gradient color bars specify the correspondence between predicted number of tracts in a given voxel and displayed colors of tracts.

in the left cortex for *D. delphis* and the right for *L. acutus* (S6 Figures A1-4, B1-4, and C1-4 in S1 File). When compared to the ascending auditory traces, IC to cerebellar traces in *D. delphis* and *L. acutus* tended to exhibit more robust paths to rostral and dorsal portions of cortex, and less prominent connections to ventral and caudal extremes (S6 Figures A1-4 and C1-4 in S1 File). For more details on these cortical projections, see S2 Text or S6 Figures in S1 File.

## Discussion

In this post-mortem diffusion tractography study of one mysticete and three odontocete brains, we replicated and extended the findings on cetacean auditory pathways from Berns et al. [41]. Further, as suggested by [42], we assessed

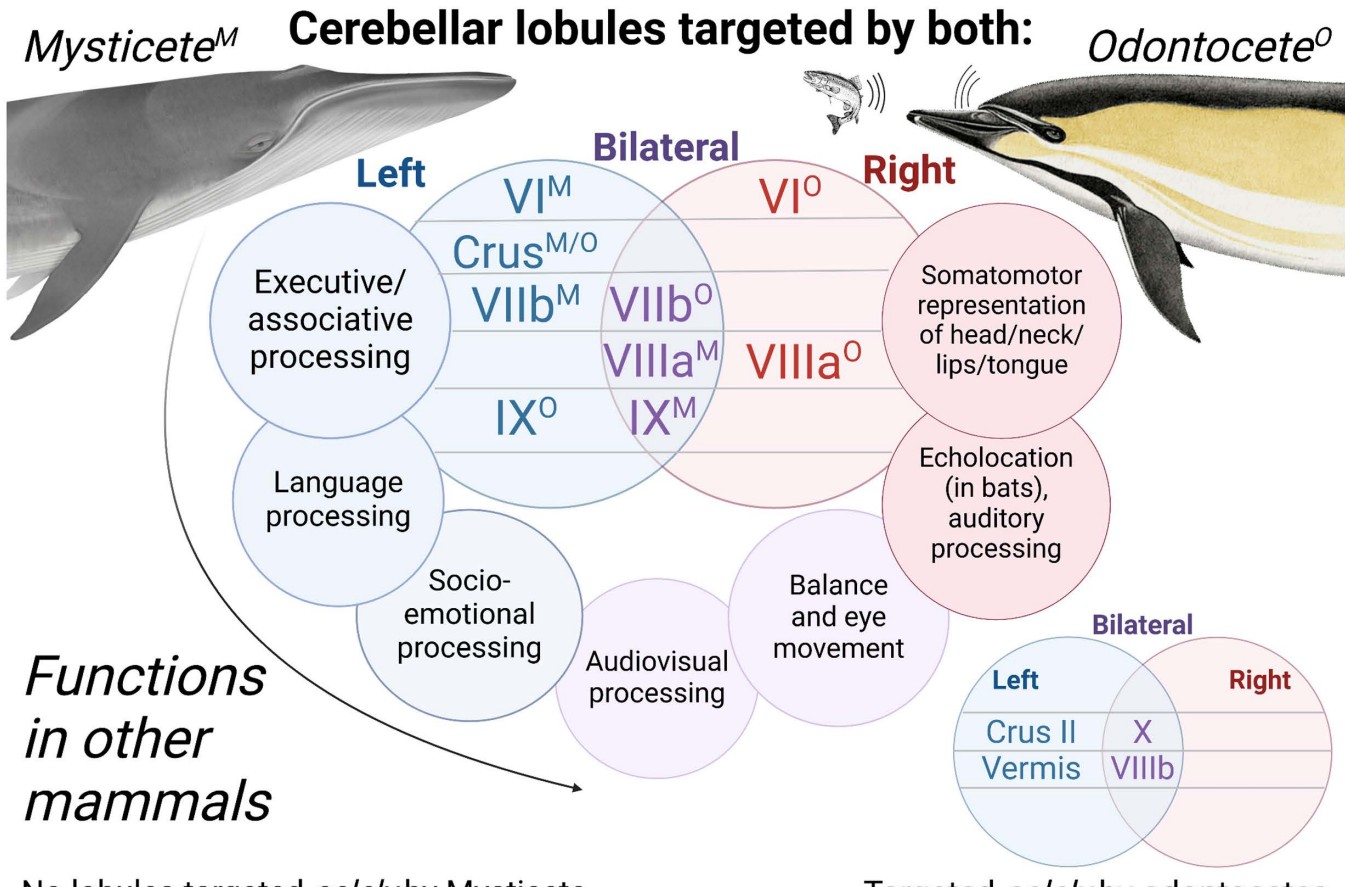

**Fig 6. Color-coded diagram displaying cerebellar lobules targeted in the left, right, or bilateral cerebella in odontocete and mysticete specimens.** Within the central venn diagram, superscript "M" signifies that a subregion was targeted in the mysticete, while "O" signifies a subregion's targeting by one or more odontocetes. Subregions that were targeted in the left cerebellar hemisphere are placed in the blue circle on the left, while right-hemisphere subregions are placed in the red circle on the right, and bilaterally-targeted subregions are placed in the overlap region colored purple. The outer ring of the central venn diagram specifies the functions associated with each lobule in other mammals [35,83–95]. Each function is placed in proximity with the lobule to which it corresponds in the inner venn diagram, and is color-coded in accordance with the subregion to which it corresponds as well. See the discussion subsection Specific cerebellar targets for more details on correspondences between subregions and functions. In the bottom corner, a smaller venn diagram displays the subregions only targeted in odontocetes, and uses the same color-coding scheme to denote whether these regions were targeted on the left side, right side, or bilaterally.

these tracts for evidence of lateralization, and found none. In all four species, tract density of ascending cortical auditory pathways (IC to cortex) was largely symmetrical. We did, however, find evidence of strong asymmetry in descending auditory pathways (IC to cerebellum) in all four species. Each of the dolphins showed greater tract density between left inferior colliculus and right cerebellum than vice versa, while the sei whale showed the opposite pattern. This lateralization of descending auditory pathways is intriguing, particularly in light of the lack of lateralization of ascending cortical tracts.

Unlike the mysticetes, which have bilateral laryngeal innervation, the odontocete sound production system separately controls the right and left phonic lips. In the dolphins, increased connectivity between left hemisphere to right cerebellum may serve to support the greater predictive sensorimotor processing demands of echolocation as opposed to passive hearing. This accords with a cluster of behavioral findings reporting that the right phonic lips more frequently produce navigational clicks in multiple odontocete species, and may support the resulting functional laterality hypothesis, which

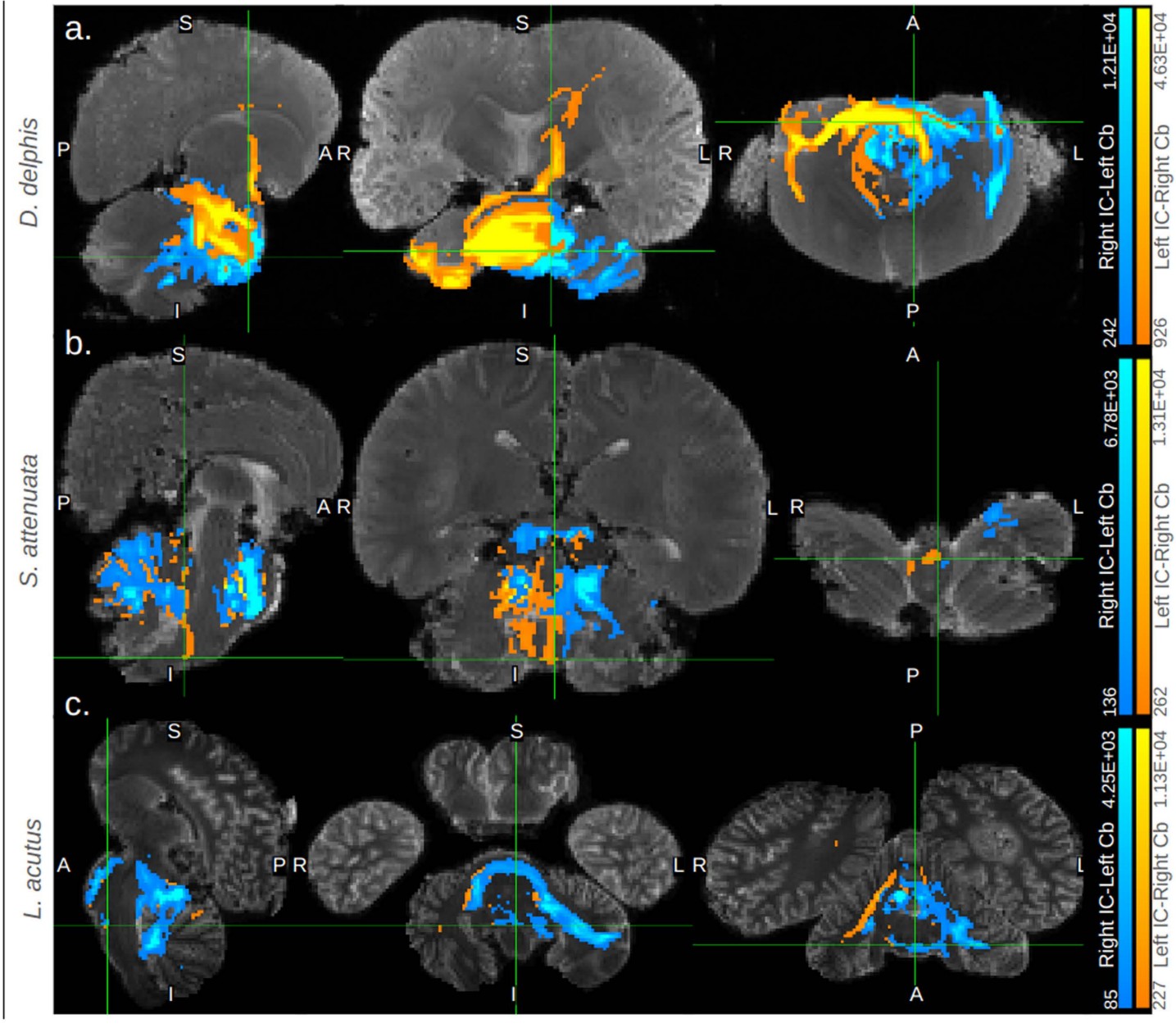

**Fig 7. Subcortical features of IC-cerebellar traces in odontocetes.** Orthographic cross-sections displaying key subcortical features of the traces between IC and cerebellum in the odontocete specimens (a) *D. delphis*, (b) *S. attenuata*, and (c) *L. acutus*. Each trace is displayed with minimum and maximum values set to 0.1% and 5% of its waytotal, respectively. Traces between right IC and left cerebellum are colored blue, while those between left IC and right cerebellum are orange. The correspondence between predicted number of tracts in a given voxel and displayed colors of tracts is given in the gradient color bars on the right edges of the panels. IC = inferior colliculi and Cb = cerebellum.

posits that echolocation clicks are preferentially produced by the right phonic lips and communication sounds by the left [52,53,79]. Although ascending auditory projections in mammals are bilateral, decussating in the brain stem, motor processing is strictly lateralized. Thus, the right phonic lips should be controlled by vocal motor regions in the left cortical hemisphere. These, in turn, would likely communicate most directly with auditory regions in the left hemisphere, which

would likely communicate most directly with the left inferior colliculus. Cerebellar-cortical communication is typically contralateral [96,97], so descending left cortical auditory signal from the auditory cortex through the IC will likely target the right cerebellum. Likewise, connections between cerebellum and motor cortex will be contralateral. Thus, we posit that the asymmetrical and robust pathways that transit left IC and right cerebellum in odontocetes likely support the rapid sensorimotor integration required for echolocation motor production, and may reflect both its volitional aspects (potentially subserved by left hemisphere auditory-to-motor cortico-cortical pathways) and its non-volitional aspects (potentially subserved by right cerebellar feedforward sensorimotor circuits) (Fig 8).

Beyond the behavioral findings described under the functional laterality hypothesis, anatomical studies of asymmetries in odontocete nasal musculature and skull also suggest that the right phonic lips may be differentially employed in the generation of echolocation clicks. In many dolphins, the fat-filled melon, which serves to direct and amplify echolocation clicks, is directly connected to the right-side, and not left-side, bursae within the phonic lip complex [81,101,102]. The right nasal air sac tends to be larger than the left one as well [103]. Further, paleontological studies of cetacean skulls from fossil records dating back millions of years suggest a general tendency toward increasing nasofacial asymmetry among odontocetes (but not mysticetes) as their echolocation capabilities have been more finely honed across successive generations [31]. These findings clearly complement the evidence at hand of an asymmetrical hypertrophy in descending auditory tracts that may inform sensorimotor integration related to vocal motor refinement in odontocetes (Table 2). Furthermore, left-side skeletal features are more likely to progressively shift in position, and nasal and melon regions responsible for sound production have shifted the most, suggesting that a progressive expansion of musculature supporting sound production on the right side has occurred at the expense of the left, leading left-side nasofacial landmarks to shift further from the median [32]. Especially when overlaid with behavioral reports of preferential use of the right phonic lips in echolocation, such paleontological reports seem to further emphasize that in our results, the robust right-side auditory-cerebellar afferents found in odontocetes may support refinement of the outgoing echolocation motor plan (Fig 8).

## Ascending auditory pathways

Qualitatively, we identified direct projections to temporal lobe from inferior colliculus in a new species of dolphin (*L. acutus*) and, for the first time, in a baleen whale (*B. borealis*). Contrary to results from early invasive electrophysiology [14,104], our findings add further support for a primary auditory field in cetacean temporal lobe, as is found in most terrestrial mammals. Further, we provide the first characterization of auditory brain projections in a large baleen species.

Despite the lack of echolocation capabilities in baleen whales, the relative density of cortical auditory tracts was greater in the sei whale compared to the three dolphin brains. Because sei whales specialize in low frequency calls [105], and high frequency auditory processing is likely to be more computationally demanding [106], it is unlikely that this difference reflects enhanced auditory processing demands in the sei whale. Despite being the oldest of the specimens, and thus the most susceptible to potential age-related fractional anisotropy (FA) loss [107], the sei whale had the highest mean FA value (Table 1). This increased FA could be due, in part, to the greater myelination required to support signal transmission over longer distances in a larger brain [108]-- though see Beaulieu [109] for a case against a specific FA–myelin correlation. If the heightened FA found in the sei whale indeed reflects greater myelination in its brain overall, it is possible that such a global tendency could affect our measures of tract strength– though this interpretation would in turn underscore the notability of our finding that descending IC to cerebellum tract strength in the sei whale was markedly lower than in the lower-FA odontocete brains (Fig 1b).

The location and organization of primary auditory cortex (A1) in odontocetes remains a mystery. Before Berns et al. [41], the general consensus among cetacean neurobiologists was that primary auditory cortex in the dolphin was posterodorsal, running along the suprasylvian gyrus adjacent to primary visual cortex ([21], although see Revishchin and Garey, [39]). While this dorsal A1 location matches the most precise odontocete electrophysiological data, it would represent a

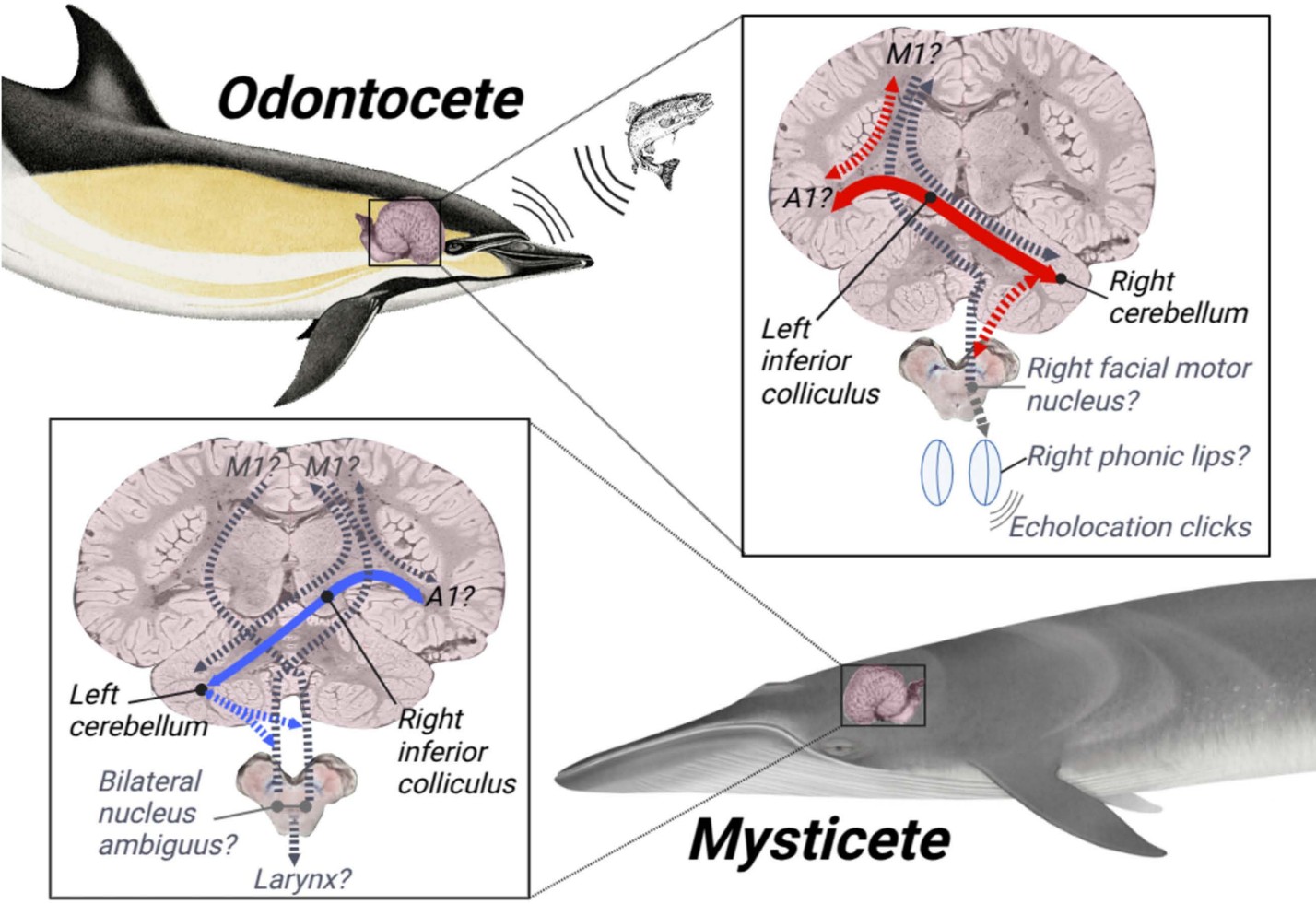

**Fig 8. Putative model of how differential auditory cortical-cerebellar connectivity may relate to differential mechanisms and functions of hearing and sound production in odontocetes versus mysticetes.** Red arrows color-code for right-cerebellar lateralization of IC-cerebellum tracts in odontocetes, while blue arrows color-code for left-cerebellar lateralization of IC-cerebellum tracts in mysticetes. The solid red arrow is thicker than the blue one to represent the higher tract strength of IC-cerebellar pathways in odontocetes. Solid red and blue arrows reflect pathways that were robustly demonstrated (i.e., withstood rigorous thresholding) in this study, while dashed red and blue arrows represent tracts that were observed more weakly (i.e., did not withstand rigorous thresholding), but may be directly or indirectly relevant to sound production pathways, and thus merit further investigation. Grey dashed arrows represent pathways that were not directly measured or observed, or could not be measured in this study (i.e., peripheral connections), but are widely conserved and/or canonical circuits between the ROIs that may be relevant to sound production. The grey dashed arrow that connects M1 and cerebellum but does *not* transit IC in the mysticete is unique in that it represents a potential takeaway from our negative finding here: since very few projections from IC-cerebellum reach putative M1 in the mysticete, it is likely that more general, widely con-served, and canonical M1-cerebellum tracts are present in these brains and involved in complex sound production [98]- though they may not be as densely connected to subcortical auditory centers as they are in echolocating odontocetes. Because our DTI tractographical method does not provide an index of directionality (i.e., efferent versus afferent, input versus output), and because our observed pathways likely involve cortico-cerebellar and cortico-cortical feedback loops, all arrows representing observed streamlines are bidirectional. Brainstem cross-sections display the hypothesized brainstem nuclei relevant for sound production in each suborder, and the hypothesized peripheral phonatory organs that would in turn receive motor signals from these nuclei are placed underneath. Coronal cortical cross-section adapted from Brain Catalogue [99], and brainstem cross-section adapted from Neurosurgical Atlas [100].

striking adaptation compared to terrestrial mammals, including the dolphin's closest terrestrial relatives, the artiodactyls, in which A1 is situated in dorsal temporal lobe [110–112]. The Berns et al. [41] findings were the first to directly support a typical mammalian dorsal temporal location for dolphin A1. We have now replicated those findings with a third dolphin species. Our parallel finding of temporal projections in the mysticete provide further evidence that the location of cetacean A1 may be conserved from the shared common terrestrial ancestor, not having diverged during the evolution of auditory mechanisms supporting echolocation following the primary odontocete split. However, it must be emphasized that our diffusion tractography results do not match the earlier electrophysiological data. There are unlikely to be new invasive electrophysiological data collected in these species, so other lines of evidence are required to address this apparent conflict. Initial work with functional near infrared spectroscopy measures suggests that, with a smaller odontocete such as a porpoise, it might be possible to access neural signal in parts of the cortex [113]. While logistically difficult, functional MRI with dolphins is also a possibility [114]. Histological analyses of putative auditory cortices may be more immediately instructive. In a recent paper examining the putative posterodorsal A1 location, [115] identified a number of unusual cytoarchitectural characteristics that do not neatly match what is known about cellular organization of A1 in terrestrial species. Notably, the region did not have the standard koniocortical organization of mammalian sensory fields. In addition, the organization of the region was less complex than the putative dolphin primary visual region. The authors suggested this might be due to some of the general whole-brain cytoarchitectural differences between odontocetes and terrestrial mammals, such as reduced or missing layer IV in dolphins and distribution of cortical processing modules over larger less densely packed layers of cortex. However, the findings are far from conclusive. Future work should at least examine cytoarchitecture in the putative dorsal temporal A1 location as well.

One possibility, raised first in [41] and echoed in [79], is that, similar to bats, dolphins may have separate cortical regions for auditory processing, including a conserved dorsal temporal region for tonotopic response and an enhanced posterodorsal cortical region for processing spatial characteristics of sound related to echolocation, including direction and delay. In the current study, lower-threshold images revealed that IC projections to cortical regions besides the dorsal temporal lobe were indeed present in the echolocating odontocetes, though less so in the non-echolocating mysticete, lending some credence to this hypothesis (S4 Figures in S1 File). However, in the current study the posterior dorsal belt of suprasylvian cortex highlighted in past literature on putative cetacean A1 was not consistently or particularly robustly implicated in IC traces across the odontocete specimens. Instead, rostro-ventral projections toward the basal ganglia (Fig 4b), dorso-medial projections toward putative cingulate and somatomotor cortices (Fig 4c), and branches to a ventrocaudal extreme of cortex (Fig 4d) were more frequently observed. Future diffusion tractography studies should further examine any putative secondary auditory projections and corticocortical connectivity with dorsal temporal lobe. In Wright et al. [42], a putative arcuate fasciculus was identified in both hemispheres with deterministic tractography. Both left and right tracts passed through the temporal lobe, but, notably, the left hemisphere fasciculus did appear to project more caudally. These tracts were extracted from whole-brain deterministic maps, however, and were not explicitly seeded in either putative A1 location.

If, as suggested by Ridgway et al. [79], dolphins do indeed privilege social auditory processing in the right cortical hemisphere and echolocation in the left, the lack of lateralization in primary ascending auditory tracts, found in in all three dolphin species and the mysticete analyzed here, is notable. Prima facie, these findings do not appear to parallel the finding of more robust arcuate fasciculus in the dolphin's right hemisphere [42]. However, corticocortical fasciculi may be lateralized even when afferent fibers targeting the relevant cortical regions are not. In addition, lateralized ascending auditory-motor connectivity, as seen in the human arcuate fasciculus, may be more relevant to long-term flexible vocal production learning, as seen in some dolphin and mysticete vocal communication [116–119], and brainstem, midbrain, and cerebellar adaptations may be more relevant to the real-time, high-speed auditory-motor integration required by echolocation. Further, although auditory motor connectivity may be enhanced in the right hemisphere to promote social communication in the dolphin, there are other potential corticocortical connections required to support

echolocation and related learning (e.g., multisensory integration to link echolocation with vision). Therefore, our findings here, of no asymmetry in ascending auditory projection strength, contextualize, as opposed to support or contradict, those in Wright et al. [41].

**Descending auditory pathways**

Although we cannot establish directionality of our tracings, we assume a priori that IC sends descending afferent auditory signals to the cerebellum, as is well established in mammalian species [59]. Here we consider the potential functional relevance of this pathway in odontocetes. Given the speed of processing required for echolocation, much has rightly been made of the raw size of early auditory nuclei in dolphin brainstem and midbrain [120–123]. Less work has examined potential contribution of the cerebellum. While transit time from auditory cortex to cerebellum might on first consideration seem too slow to meaningfully contribute to echolocation, the cerebellum appears to operate as a feed-forward Bayesian system for sensorimotor integration and related movement planning and refinement [65,72,124]. Part of what the cerebellum probabilistically models is the organism's own subsequent behavior based on the current sensorimotor situation, leaving open the possibility for a strong cerebellar role in both click production and body-movement based on click production and reception. In Bayesian terms, the dolphin's current sensorimotor situation, comprising body location and movement, vocal motor echolocatory production, and echoic feedback could be integrated with prior knowledge of successful and unsuccessful movements and click productions in similar situations, with the potential to output a feedforward motor plan judged most likely to be successful for the immediate motor goal. This is very much in line with evidence and interpretation from Beedholm et al. [125], which showed that evoked auditory responses to echolocation clicks can happen *after* subsequent echolocation clicks are produced. That is, an echolocating dolphin often has to produce a new outgoing click with a set time/intensity before having been able to process the returning output from the most recent click. A cerebellar model of sensorimotor prediction has also been proposed for general mammalian hearing, wherein auditory stimuli are expected as a byproduct of an upcoming motor act [126].

Although data on the functional role of cerebellum in dolphin echolocation are limited, there is strong evidence of cerebellar hypertrophy in the odontocete line [73,83]. In the present study, the sei whale cerebellum had similar relative brain volume as the Atlantic white-sided dolphin, but the common dolphin and pantropical spotted dolphin cerebella were notably larger (Table 1). Prior evidence indicates that baleen whales and also sperm whales (which produce slower clicks) have smaller relative cerebella in comparison to most toothed whales [6,127]. It appears that the cerebellum does not scale allometrically with brain size in cetaceans. Given the apparent enlargement of auditory regions in the cerebella of echolocating odontocetes, Oelschlager [21] has suggested there may be a link between cerebellar hypertrophy and echolocation. This would parallel extensive data showing A. relatively larger cerebella in bats compared to non-echolocating bats and terrestrial rodents [128], and B. the specific physiological involvement of the bat cerebellum in echolocatory processing [77,129–131]. Specifically, auditory receptive regions in bat cerebellum have been found to code for range and direction of moving targets, which is likely necessary for the rapid, subcortical feedforward motor decisions required to get where a target is going, as opposed to where it was.

Given the functional laterality hypothesis and its prediction of lateralized neural processing of production and reception of communication vs. echolocation sounds, the fact that primary ascending cortical auditory tracts in the dolphins were not as heavily lateralized as the descending tracts tentatively suggests that, at least in terms of post-IC neural processing, the cerebellum may play a larger role in managing echolocation-specific computation than the literature has previously suggested. In baleen whales, which do not appear to have lateralized sound production mechanisms [80], it remains to be seen to what extent the functional laterality hypothesis applies, if at all. In the current study, the lateralization pattern of auditory cerebellar pathways was reversed in the sei whale in comparison to the dolphins, with far denser tracts entering left cerebellum than right. However, the fact that *left* tract density was roughly the same in all four species suggests that *hypertrophy* of the right tract, and *not* a reduction of left-cerebellar auditory processing, may drive the asymmetry

seen in the dolphins. Why, then, is the sei whale's right-to-left descending auditory tract so much stronger than its right? Baleen whales, just like odontocetes, show right side bias for social and feeding behavior, which has been suggested to be rooted in left-hemisphere sensory biases [132,133], so the sei whale's left-cerebellar lateralization is unlikely to be due to primary sensorimotor processing. The cerebellum is a multipurpose structure, and serves many divergent and integrative functions. One possible explanation is that, if, as in the dolphins, the right cortical hemisphere is specialized for communication in baleen whales, the left cerebellum may play an important role in production and refinement of vocal communication signals. There are bird data showing the importance of the cerebellum to vocal learning [134], and there is evidence in humans that the cerebellum contributes meaningfully to language processing [93,135]. Across a number of non-echolocating species, new evidence indicates direct cerebellar connectivity with phonatory brainstem nuclei [136]. It is plausible that a right cortical-left cerebellar circuit for social vocal learning is present in both toothed and baleen whales. One interpretation of our current findings is that while all whales, toothed and baleen, have complex social vocal behavior that is right-hemisphere lateralized with robust support from descending auditory pathways to the left cerebellum, toothed whales have also added a hypertrophic descending pathway from the left cortical hemisphere to the right cerebellum to manage the processing-intensive, highly time-constrained demands of echolocation. The question remains whether baleen whales do show right-lateralized vocal communication processing at the cortical level. A number of terrestrial species have been found to have left-lateralized vocalization mechanisms, but preliminary findings in hoofed mammals do not clearly support this pattern [137], leaving open the possibility that right-lateralized social vocal mechanisms are conserved in cetaceans from a terrestrial ancestor.

## Specific cerebellar targets

Existing data on the function of specific cerebellar lobules in other mammals can be tentatively extrapolated to support our conjectures regarding the strength and laterality of ascending and descending auditory pathways in cetaceans. Extensive physiological and anatomical evidence suggests that the posterior and lateral lobules of the cerebellum have evolved to support highly skilled behavior and domain-general cognitive abilities in humans and multiple distinct mammalian lineages [98,138]. Meanwhile, medial and anterior cerebellar regions are hypothesized to subserve more traditionally-noted cerebellar functions like sensorimotor representation and limb coordination [138,139]. In humans, prefrontal (PFC) and parietal associative areas tend to coactivate with posterior and lateral areas, *particularly* Crus I and II, with Crus I being especially linked to PFC, executive function, and working memory tasks, and Crus II being especially linked to PFC *and* parietal association areas, social cognition, and theory of mind [89,92,94]. Meanwhile, motor cortex tends to coactivate with anterior lobules plus lobule VIII [89,92], and the vermis appears to be most active in tasks involving socio-emotional processing [139]. Interestingly, language processing appears to involve Crus I and II [92], with Crus I being more linked to syntactic and Crus II to semantic processing [93], and generally, comparative evidence supports the idea that vocal learning and lateral cerebellar expansion are linked in mammalian lineages [98]. The apparent targeting of *left*-lobule Crus I and II by right IC-left cerebellar traces in the odontocetes seems to accord with the putative role of the *left* phonic lips in communication and social sound production in odontocetes [52,53,79]. This interpretation is further backed by the targeting of the vermis, also linked to socio-emotional processing [91], by the right IC-left cerebellar traces in two out of the three odontocetes.

Notably, compared to humans, Crus I and II are fairly diminutive in cetaceans [35], while lobule IX, VIIIa and b, VIIb, and VI are greatly expanded [83]. Existing mammalian findings help make sense of the pathways observed in these lobes in the current study. Suga and Horikawa [88] found that lobules IX and VIIIa/VIIIb are active in bats during production of echolocation sounds, and additional studies in humans, felines, and other mammals suggest lobules VIIIa/VIIIb are related to audition, auditory motion processing, audiovisual integration, and sensorimotor representation [69,92]. The targeting of *right* VIIIa in odontocetes, but bilateral VIIIa in the mysticete, is suggestive of this study's core line of interpretation: namely, that left IC to right cerebellar pathways have proliferated to subserve echolocation and acousticomotor integration

in odontocetes, while *non-social* acousticomotor tracts have remained bilateralized and are less developed in the mysticete. Furthermore, IC to lobule VI pathways were more right-dominant in the odontocetes, but were biased toward left cerebellum in the mysticete. Lobule VI has been linked to audition as well as somatosensory and motor representation in face, neck, head, tongue, and lips [35,69,83]. In odontocetes, the left IC to right cerebellar tracts' presence in lobule VI seems to align well with the proposed role of these tracts in echolocation, given that echolocation involves tight integrative looping of somatosensory information from the head with precise motor coordination of their musculature [15,35]. It should be noted that Hanson et al. [83] found that lobule VI was larger on the left than the right in *Tursiops*. The size disparity might be driven by non-auditory processing related to sound production by left phonic structures (i.e., whistling) and patterns of non-IC connectivity from these lobules should be examined in toothed and baleen whales.

In sum, the observation of qualitatively pronounced tracts between right IC and left Crus I and II in odontocetes corroborates the hypothesized specialization of their right cortex and left cerebellum for social and communicative functions. Importantly, because Crus I and II are quite small in cetaceans [35,83], tracts to these regions may not be as quantitatively robust as those to other lobules, and this helps make sense of the comparatively weaker tract strength observed in odontocete right IC to left cerebellar traces. Meanwhile, IC projections to lobules involved in sensorimotor representation, and especially in audition or head, neck, and face somatomotor activity, appear to be more pronounced between the left IC and right cerebellum in odontocetes, and this is the side that is *quantitatively* more robust in the odontocetes too. As mentioned previously, dolphin trigeminal and facial nerves heavily innervate the phonic lips and associated sound-production mechanisms. The qualitative-quantitative accordances here strongly support the hypothesis that right cortical to left cerebellum tracts support social and communicative functions across all cetaceans, but that in odontocetes, additional tracts have asymmetrically proliferated to connect the left cortex and right cerebellum to support the rapid acousticomotor integration demands of echolocation.

On the other hand, our cerebellar projection findings contain numerous remaining mysteries. Lobule IX is the largest in cetaceans but fairly small in humans, though it is unclear precisely what role it plays in cetaceans [35,83]. In rodents, this lobule seems to be involved in auditory *and* visual processing [85,87,140]; in bats, it is again linked to production of echolocation sounds [88]; and in humans, it may play a role similar to Crus I/II in language and social cognition [92], or may be linked to auditory processing as well [141]. Lobule IX's expansion in cetacea therefore could be linked to an expansion of sensorimotor feedback and audio-visual integration capabilities for echolocation, but it may well serve as a substrate for expanded social and communicative behavior, too. The left-side prominence of IC to cerebellar tracts to this lobule in odontocetes, in light of the other findings at hand, tentatively suggests the latter. Finally, lobule VIIb, while fairly large in cetaceans [83], is acutely understudied, rendering it difficult to postulate its functional significance in cetacea. The minimal literature that does exist suggests it might be involved in visuospatial working memory in humans [86]. Some older evidence in cats indicates its potential role in audition and sensorimotor limb representation [84]. Though interpretations are limited by the sparseness of existing evidence, is possible that in cetaceans, this lobule may be involved in sensorimotor representation of the body stem and fins [35], and perhaps even in the continuous integration of spatial, motor, and proprioceptive information about the body and its motion within the environment.

### Contralateral ascending pathways

Our data pose a further mystery. While direct ipsilateral ascending tracts from IC to cortex were generally not lateralized, contralateral projections (from right IC to left cortex and from left IC to right cortex) *were* lateralized, with denser tracts going from left IC to right cortex. The exception to this was *D. delphis*, in which ipsilateral ascending tracts were more robust between right IC and right cortex than left IC and left cortex, and contralateral projections between IC and cortex were non-lateralized. The cerebella were necessarily excluded for all specimens in both ipsilateral and contralateral cortical traces used for the quantification of ascending auditory tract strength so that streamline counts did not include IC to cerebellar tracts. In follow-up traces of ipsilateral IC connectivity *without* cerebellar exclusion conditions in *D. delphis*, a

less robust lateralization effect was observed, suggesting the seemingly robust right-lateralization initially discovered may in fact partially reflect a *relatively greater loss* of left-side IC tracts and a *relative lack of loss* of right-side connectivity in *D. delphis* upon the exclusion of contralateral and cerebellar connections (S8 Table 1 in S1 File). Similarly, follow-up traces in *L. acutus* revealed that removing cerebellar exclusion conditions in the contralateral IC-cortical traces attenuated the right-lateralization effect, suggesting that the right-lateralization effect may be partially mediated by a relative *lack of loss* of right-side IC connectivity due to the exclusion of the cerebella (S8 Table 3 in S1 File). These findings tentatively reinforce the notion that quantitatively robust cerebellar connectivity is more relevant to the left IC and cortex than the right in the odontocetes. It may be that some significant portion of auditory cortical projections to the left hemisphere in dolphins are reciprocally connected with the cerebellum. Such pronounced integration between left auditory cortex and right cerebellum, in light of the functional laterality hypothesis, further suggests this circuit's role in the production and reception of echolocation sounds.

A separate interpretation of the finding of right-cortical bias for contralateral IC-cortical connectivity is that auditory information from both ears is preferentially sent to the hemisphere more responsible for processing communication, as opposed to echolocation. One possible explanation is that this represents segregation of echolocation sound processing to one hemisphere to prevent interference from communicative signals, as has been suggested in bats [142], or, more broadly, could reflect the specialization of different cortical regions for different types of neural processing. This may be useful given that dolphins employ high frequency pulses, produced by the left phonic lips, in communicative contexts, signals that have some spectral overlap with echolocation vocalizations [52,143]. Further, communication and echolocation signals can be produced simultaneously [79]. However, there may be other factors at play, such as asymmetry in sound receptivity of external hearing pathways on each side of the dolphin head [144]. Lateralization of sound production and hearing in baleen whales is understudied, but if they are right-hemisphere lateralized for communicative sound production, it may be more efficient to get auditory information to this hemisphere as opposed to the left hemisphere for this species. There is a tendency toward right-side behavioral lateralization in cetaceans as well, with related perceptual adaptations [133]. Regardless of laterality, in terms of overall tract strength, interhemispheric connectivity in cetaceans is generally low [42], and our findings reflect this as well, as in all specimens, the strength of contralateral IC-cortical tracts were fairly weak compared to ipsilateral cortical projections.

## Limitations

In the current study, our findings were limited predominantly by sample size– it has been particularly difficult to acquire usable, preserved whole baleen whales for tractography, due to both regulatory concerns and the logistical demands of acquiring, fixing, and imaging such large brains. Further collaboration between scientists and veterinary and stranding response personnel may provide more opportunities for acquiring these valuable specimens and data.

As with all diffusion tractography studies, there are known limitations to the accuracy of the tensor fit and pathway construction [145]. However, our data were acquired with a specialized SSFP imaging sequence optimized for post-mortem tissue [43,146]. This approach allows for high-resolution (sub mm isotropic), high signal-to-noise ratio imaging, with a large number of sampled angles (52 in our case), and makes use of the FSL crossing fibers model [147], which, together, limits some of the concerns regarding anatomical precision. Further, the preprocessing involves specifically modeling and accounting for divergent T1 and T2 tissue values, which helps control for accumulating changes to tract integrity and diffusion parameters with increased time post-mortem. The primary tracts we assessed in the current study were very robust, even with heavy thresholding.

## Future work

The current findings would be complemented by a parallel study with a sperm whale brain. Sperm whales have the smallest relative cerebellum size among the odontocetes [127]. In addition, they have only *one* pair of phonic lips, and

these are situated on the right side of their head, where they are presumably employed for both social and echolocation sound production [148]. It is of great interest whether sperm whales also show right cerebellum lateralization of auditory processing.

The hypotheses put forth above also assume integration of auditory and vocal motor processing in cetaceans. However, there have been essentially no studies of vocal motor pathways in the cetacean brain. Their apparent capacity for vocal production learning and the relevance of vocal production to auditory processing in echolocation makes this a tempting target [117,149,150]. Although vocal motor regions of the cetacean *cortex* have not been identified, relevant brainstem nuclei have been mapped via prior histological work and could be used to seed tractography [35].

## Conclusion

In summary, here we have extended the diffusion tractography evidence on the odontocete auditory system, and have presented the first diffusion MRI data in a baleen species to wit, allowing a landmark comparison of auditory tracts in echolocating versus non-echolocating whales. Each brain showed, as in Berns et al. [41], a clear pathway from IC to temporal lobe via the thalamus, further challenging the account that primary auditory processing has been completely dorsally-shifted in odontocetes. The most striking difference between the species was in the density and lateralization of descending IC to cerebellum tracts, with notable hypertrophy in right-cerebellar tracts in the odontocetes. Given behavioral evidence that the right phonic lips are preferentially employed to produce echolocation clicks in some odontocetes, this asymmetrical hypertrophy may reflect a heightened demand on the right cerebellum to perform the rapid, precise, and predictive sensorimotor processing likely required for echolocation. Meanwhile, density and lateralization of ascending auditory tracts did not differ significantly between the echolocating and non-echolocating species. We suggest that the specialization of auditory-cerebellar processing for echolocation in odontocetes may represent parallel evolution to that seen in echolocating bats. This heretofore underexplored neural component of the cetacean echolocation system should be more closely examined in future studies, including diffusion tractography, histology, behavioral, and noninvasive electrophysiological methods.

## Materials and methods

### Specimens

In total, brains from four different cetaceans were used in this study, including three odontocetes and one mysticete. The first specimen was a post-mortem brain extracted from a pregnant adult female common dolphin (*Delphinus delphis*) that stranded dead in Buxton, North Carolina (Field #PTM135) in February 2001 [41]. The second specimen was a post-mortem brain extracted from an adult female pantropical spotted dolphin (*Stenella attenuata*) that stranded dead at Camp Lejeune, North Carolina (WAM576). Both carcasses were in fresh condition (Smithsonian Condition Code 2 [151]) with no apparent damage, and the brains were extracted while still fresh. The total body length and body weight of the *D. delphis* and *S. attenuata* were 203 cm and 191 cm, and 83 kg and 57 kg, respectively. The brain of *D. delphis* had a fresh weight of 981 g, an anterior-posterior length of 132 mm, a bitemporal width of 155 mm, and a height of 96 mm. Up until the scanning of the *D. delphis* brain in Marino et al. [19] and the scanning of both brains in Berns et al., [41] both specimens were stored at Emory University, where they were kept in 10% neutral buffered formalin that was changed regularly. The images procured during the scanning session for Berns et al. [41] are used in the current study. Thirty minutes prior to the Berns et al. [41] scanning session, the specimens were set in 2% agarose (Phenix Research Products Low EEO Molecular Biology Grade Agarose) doped with an insoluble mixture of 2 mM gadolinium (III) oxide (Acros Organics, Fisher Scientific) to minimize artifacts and enhance tissue contrast (D'Arceuil et al., 2007; Miller et al., 2011) [43,152].

The third specimen was a post-mortem brain extracted from an adult female Atlantic white sided dolphin (*Lagenorhyncus acutus;* specimen code CCSN03–164La; Smithsonian Condition Code 2 [151]) that was found dead in Cape Cod, Massachusetts in 2003. A necropsy was performed the day following the animal's discovery, and the extracted brain was

fixed in 10% phosphate-buffered formalin and stored at Woods Hole Oceanographic Institute until the scanning session described below. Histological analysis determined the animal had an extensive fungal infection with degeneration, necrosis, and inflammation at the time of death; however, there was no evidence that this infection damaged the brain. The total body length of the animal was 213.5 cm.

The fourth specimen was a post-mortem brain extracted from an adult sei whale (*Balaenoptera borealis*), sex and body length unknown. Although the whale was initially killed for its meat by Icelandic hunters in August of 1984, the scientific use and transportation of its tissues was approved by the Icelandic and United States governments. Its brain was extracted while the carcass was fresh (Smithsonian Condition Code 2 [151]) and then stored in formalin, where it remained for decades. At some point, a caudal piece of cortex from the right temporal lobe was sampled for histological examination; fortunately, the sampling was relatively shallow and likely did not disrupt the integrity of white matter tracts. Outside of the minor damage created by this sampling, the brain was found to be in remarkably good condition prior to its scanning in the 3 Tesla Siemens Trio MRI at University of California, Berkeley, as will be further described below.

## Imaging

As described in Berns et al. [41], the brains of *D. delphis* and *S. attenuata* were scanned in a 3 T Siemens Trio using standard gradients (40 mT/m maximum) and a 32-channel head receive coil over the course of approximately 8 hours. Following Miller et al. [43], DTI was carried out using a diffusion-weighted steady state free procession (DW-SSFP) sequence, as this acquisition method is preferable for maximizing the signal-to-noise ratio in diffusion-weighted images of post-mortem brains. For each brain, a set of DW-SSFP images were collected, weighted along 52 directions (FOV = 166 mm, voxel size = 1.3 mm isotropic, TR = 31 ms, TE = 24 ms, flip angle = 29°, bandwidth = 159 Hz pixel$^{-1}$, $q$ = 255 cm$^{-1}$, $G_{max}$ = 38 mT m$^{-1}$, gradient duration = 15.76 ms [42]). Additionally, six images were collected to serve as a signal reference, and their acquisition followed the above parameters, except with $q$ = 10 cm$^{-1}$ solely applied in one direction. Because the progressive relaxation of post-mortem brain tissue leads its T1 and T2 values to differ significantly from those observed in in vivo tissue, we calculated these values from a series of T1- and T2-weighted images to ensure proper modeling of the DW-SSFP signal. The T1-weighted images were acquired with a TIR sequence in which TR = 1000 ms, TE = 12 ms, and TI = 30, 120, and 900 ms, while T2-weighted images were acquired with a TSE sequence in which TR = 1000 ms, TE = 14, 29, and 43 ms. Structural image acquisition was performed using a balanced SSFP sequence (TR = 7.03 ms, TE = 3.52 ms, and flip angle = 37°), and balanced SSFP images were collected in pairs with the RF phase incrementing 0° and 180°, which were later averaged to minimize banding artifacts [43]. The resulting structural images had a high resolution of 0.6 x 0.6 x 0.5 mm.

The *L. acutus* and *B. borealis* brains were scanned at the Brain Imaging Center of the University of California, Berkeley. The brains were again scanned in a 3 T Siemens Trio with standard gradients, a 32-channel head receive coil, and the DW-SSFP scanning protocol adapted from Miller et al. [43]. Because of its size, the *B. borealis* brain was held only in plastic Ziploc bags to prevent drying, and was inserted intact-hemisphere-first into the 32-channel coil. Meanwhile, the *L. acutus* brain was held in a plastic Ziploc bag filled with perfluoropolyether (PFPE) fluid, and was inserted rostral-first into the scanner with the cerebellum to the rear. Trying to hold the buoyant specimen fully under the fluid proved difficult, resulting in about 2/3 coverage. As a result, some fixative floating on the top of the PFPE is seen as a rim around the $q$ = 20 images. As above, images were collected along 52 diffusion weighting directions, though this time a maximum of 8 diffusion weighting directions per block was established due to file size limitations on the acquisition computer. Accordingly, acquisition occurred in blocks of (3,8) = 11, (5,7) = 12, (3,5,6) = 14, and (7,8) = 15 directions. Extra $q$ = 20 scans were included between every block to account for the long acquisition times. For *B. borealis*, FOV = 230 mm x 230 mm, voxel size = 1.1 mm$^3$, TR = 34 ms, TE = 25 ms, flip angle = 44°, bandwidth = 90 Hz pixel$^{-1}$, $q$ = 270 cm$^{-1}$, $G_{max}$ = 38 mT m$^{-1}$, and gradient duration = 16.7 ms. Each individual DW-SSFP scan took 15 minutes and 14 seconds, for a total acquisition time of 16 hours. For *L. acutus*, FOV = 112 mm x 190 mm, voxel size = 1 mm$^3$, TR = 34 ms, TE = 25 ms, flip angle = 34°, bandwidth = 90 Hz pixel$^{-1}$, $q$ = 269 cm$^{-1}$, $G_{max}$ = 38 mT m$^{-1}$,

and gradient duration = 16.7 ms. Each individual DW-SSFP scan took 9 minutes and 10 seconds, for a total acquisition time of 9 hours. Directed bore ventilation was used to maintain the specimens at a uniform temperature.

## Processing

Following image acquisition but prior to tractographical analysis, images were processed by adapting FSL tools to suit the DW-SSFP signal model. The processing method described in Berns et al. [41] for the *D. delphis* and *S. attenuata* specimens was repeated here with the new specimens of *L. acutus* and *B. borealis*. To approximate the T1 and T2 values of the brains, we assessed the image intensities of several T1 and T2 images with different TR and TE values. The T1 and T2 values of all brains were estimated to be 350 and 50 ms, respectively, yielding an approximate $b_{eff}$ value of 3500 s mm$^{-2}$ in the diffusion-weighted scans [146]. As before, all diffusion images were registered to a common q = 10 cm$^{-1}$ reference image, and all reference images averaged to create a mean reference image. The diffusion tensor model for describing anisotropic diffusion was fit to each voxel using a Metropolis Hastings algorithm that imposes a positivity constraint on tensor eigenvalues. This procedure produced estimates for the fractional anisotropy, mean diffusivity, and three eigenvectors of the diffusion tensors. Finally, a modified BedpostX model with DW-SSFP signal equations was applied with default settings, save for the specification of 2 crossing fibers per voxel [43,146,153].

## ROI selection and tractography

Following Berns et al. [41], the inferior colliculi (IC) were selected as the key region of interest (ROI) for the auditory pathway in all specimens. The IC were selected as the start point because nearly all ascending auditory information passes through them [35], and also because they are large, prominent, and easy to identify. Additionally, diffusion-weighted imaging has been used to plausibly trace IC tracts in humans [154]. For reference, in the following, all referenced sagittal planes span between the lateral-most points of each hemisphere; all coronal planes span from the rostral-most point of the cerebral cortex to the caudal-most point of the cerebral cortex; and all axial planes span from the dorsal-most vertex of the cortex to the ventral-most point of the brainstem. In other words, analyzed brains are presented in orientations akin to those employed for human brains, rather than the more naturalistic 90-degree-rotated orientation that reflects the actual positioning of the brain in the skull of cetaceans in vivo. In all specimens, the IC were identified as round protrusions on the caudal surface of the midbrain, apparent in the sagittal view dorso-rostral to the cerebellum and ventro-caudal to the superior colliculi and thalamus; apparent in the coronal view dorsal to the lateral lemniscus and ventral to the thalamus; and apparent in the axial view rostral to the ventral extremity of the cortical hemispheres, and caudal to the dorsal extremity of the pons. For the reasons described previously, the cerebellar hemispheres were deemed instrumental for the study of descending acousticomotor pathways, and were designated as the second ROI. The cerebellar hemispheres were also easily identified by their large size and unmistakable gross anatomical structure, protruding from the caudal surface of the midbrain and brainstem, just ventro-caudal to the inferior colliculi.

ROIs used to seed tractography were created as masks over high-resolution T2-weighted structural images in FSLeyes. In the present study, all voxels in each inferior colliculus and cerebellar hemisphere (left and right) were selected in accordance with the above-listed anatomical guidelines (Fig 9). Additionally, single-voxel-wide whole-plane masks were drawn at the medial-most sagittal plane between left and right hemispheres in order to examine and quantify tracts that were strictly contralateral and strictly ipsilateral. A single-voxel-height mask of an axial slice of the pons and upper brainstem was also created to allow specific analysis of tracts that ascended beyond this axial plane to the diencephalon and telencephalon, and to exclude tracts that did not pass to a level more dorsal than the metencephalon and myelencephalon. The masks were then loaded with the BedpostX file into the FSL tract-tracing tool, ProbtrackX, in order to perform the probabilistic tracing. Tracking was performed with the default settings of 5000 samples per voxel, 0.2 curvature threshold, and 0.5mm step length. The sole change made to ProbtrackX default settings was the enabling of the distance correction setting, given the large size of the brains being analyzed. To allow specific and isolated analysis of the potential

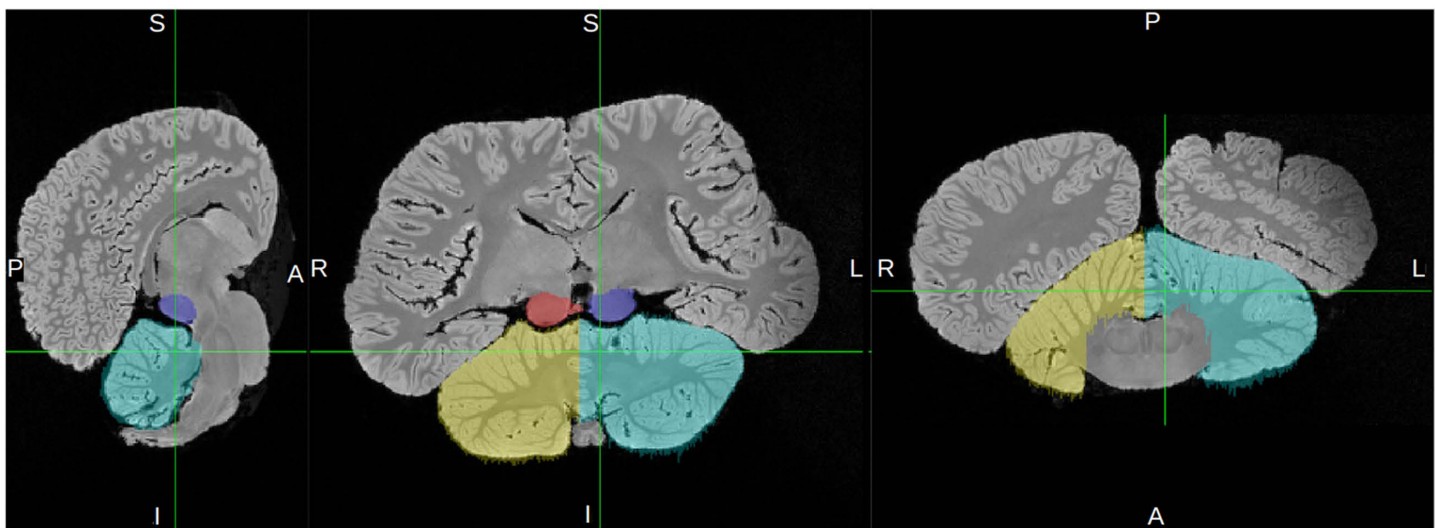

**Fig 9. Inferior Colliculi and Cerebellum ROIs.** Sagittal, coronal, and axial cross-sections showing the masks made for the cerebella (turquoise and yellow for left and right, respectively) and inferior colliculi (blue and red for left and right, respectively) in *B. borealis*. Refer to S3 Figures in S1 File to view these masked ROIs in the other specimens.

transmission of auditory information via each IC to ipsilateral and/or contralateral cortices and cerebella, we conducted a systematic battery of traces in which traversal of at least one voxel in a cerebellar hemisphere and/or the contralateral side of the brain was either necessary for inclusion of a tract (i.e., mask files were added as ProbtrackX waypoints) or grounds for exclusion of a tract (i.e., mask files were added as ProbtrackX exclusion masks). The full battery of traces, performed with every possible permutation of inclusion/exclusion criteria, can be viewed in S8 Tables in S1 File.

Following tractography, we quantified the output using *waytotal* values: the count of streamlines that satisfy the tractography algorithm by originating in the seed ROI and satisfying waypoint and/or exclusion criteria [155]. Additionally, volumetrics for each animal and each ROI were calculated by dividing the volume of each masked ROI by its specimen's whole-brain volume. Notably, the volumetric values for each whole brain and region may not precisely reflect the true volumes in live animals, as formalin fixation can lead to marked tissue shrinkage [127]-- though Cook et al. [156] and Montie et al. [157] present evidence to the contrary. Additionally, though differences in fixation times may cause variations in degrees of shrinkage *between* specimens, to wit, there is no substantial evidence suggesting disproportionate shrinkage occurs between different regions, suggesting that calculating the *relative* volumes of regions within formalin-fixed brains remains valid. For each diffusion tractography output, i.e.,"trace," these whole-brain volume measurements were used to calculate tract *strength* by dividing the trace's waytotal by the volume of the whole respective brain. Tract *lateralization* was calculated first by simply dividing the larger waytotal by the smaller one (ex. right IC trace waytotal divided by left IC trace waytotal) to provide the *factor* by which each trace was left- or right-lateralized. Additionally, the lateralization index (LI) for each trace was calculated by applying the LI formula used in Vernooji et al. [82] and Wright et al. [42], (left - right)/(left+right), to the left and right waytotal values for each trace. Finally, waytotal values also provided image thresholding guidelines for qualitative analysis, as discussed below.

## Thresholding

All tract images in this paper are presented at one of two thresholds: a more conservative one optimized for the FSLeyes orthographic view and a more liberal one optimized for the FSLeyes three-dimensional (3D) view. For the orthographic

images, traces were thresholded with a minimum value set to 1% of their waytotal and a maximum value set to 30% of their waytotal. Because the 3D view in FSLeyes tends to render tractograms more weakly than in the orthographic view, in the interest of conferring useful and detailed visual information about the sites of tract projection, we selected a more liberal threshold for tracts in the 3D view. For these images, traces were thresholded with a minimum value set to 0.10% of their waytotal, and a maximum value set to 5% of their waytotal. See the highlighted rows in S8 Tables in S1 File for the raw waytotal values that were used to calculate these thresholds.

## Ethics declaration

We affirm that this work involved no commercial incentives that could create potential conflicts of interest. Additionally, we affirm that no animals were harmed in this research. All tissue was obtained from opportunistically-collected post-mortem specimens and in accordance with both the Marine Mammal Protection Act and the Guideline for the Treatment of Marine Mammals in Field Research as established by the Society for Marine Mammalogy [158]. The collection of the *Lagenorhynchus acutus* brain used in this study was conducted under a federal stranding agreement between IFAW and NOAA under the Marine Mammal Protection Act, as is the case for all IFAW activities related to strandings. The *Stenella attenuata* and *Delphinus delphis* tissues were extracted and transported with the support of organizations including the University of North Carolina Wilmington's Marine Mammal Stranding Program and the National Marine Fisheries. Additionally, the international transport of the *Balaenoptera borealis* brain was approved by the U.S. Customs and Border Protection, Icelandair, and the Republic of Iceland.

## Supporting information

**S1 File.** **S1 Text.** Detailed cerebellar and subcortical projection sites in IC-cerebellar traces. **S2 Text.** Detailed cortical projections in IC-cerebellar traces. **S3 Figures.** Masked regions of interest in FSLeyes. **S3 Figure A.** D. delphis. Red= right, blue=left for cerebella, yellow=right and turquoise=left for inferior colliculi. **S3 Figure B.** S. Attenuata. Red= right, blue=left for cerebella, yellow=right and turquoise=left for inferior colliculi. **S3 Figure C.** L. acutus. Red= right, blue=left for cerebella, yellow=right and turquoise=left for inferior colliculi. **S3 Figure D.** B. borealis. Red= right, blue=left for cerebella, yellow=right and turquoise=left for inferior colliculi. **S4 Figures.** Ascending auditory tractograms. **S4 Figure A1:** D. delphis, left IC tracts shown in blue, right IC tracts shown in red, minimum threshold set to 1% and maximum threshold set to 30% of waytotals. Orthographic view. **S4 Figure A2:** D. delphis, left IC tracts shown in blue, right IC tracts shown in red, set to a more liberal threshold of minimum 0.1% and maximum 5% of waytotals. Orthographic view. **S4 Figure A3:** D. delphis, left IC tracts shown in blue, right IC tracts shown in red, set to a more liberal threshold of minimum 0.1% and maximum 5% of waytotals. Still 3-dimensional view. **S4 Figure A4:** D. delphis, left IC tracts shown in blue, right IC tracts shown in red, set to a more liberal threshold of minimum 0.1% and maximum 5% of waytotals. Rotating 3-dimensional view. **S4 Figure B1:** S. attenuata, left IC tracts shown in blue, right IC tracts shown in red, minimum threshold set to 1% and maximum threshold set to 30% of waytotals. Orthographic view. **S4 Figure B2:** S. attenuata, left IC tracts shown in blue, right IC tracts shown in red, set to a more liberal threshold of minimum 0.1% and maximum 5% of waytotals. Orthographic view. **S4 Figure B3:** S. attenuata, left IC tracts shown in blue, right IC tracts shown in red, set to a more liberal threshold of minimum 0.1% and maximum 5% of waytotals. Still 3-dimensional view. **S4 Figure B4:** S. attenuata, left IC tracts shown in blue, right IC tracts shown in red, set to a more liberal threshold of minimum 0.1% and maximum 5% of waytotals. Rotating 3-dimensional view. **S4 Figure C1:** L. acutus, left IC tracts shown in blue, right IC tracts shown in red, minimum threshold set to 1% and maximum threshold set to 30% of waytotals. Orthographic view. **S4 Figure C2:** L. acutus, left IC tracts shown in blue, right IC tracts shown in red, set to a more liberal threshold of minimum 0.1% and maximum 5% of waytotals. Orthographic view. **S4 Figure C3:** L. acutus, left IC tracts shown in blue, right IC tracts shown in red, set to a more liberal threshold of minimum 0.1% and maximum 5% of waytotals. Still 3-dimensional view. **S4 Figure C4:** L. acutus, left IC tracts shown in blue, right IC tracts shown in red, set to a more liberal threshold of minimum

0.1% and maximum 5% of waytotals. Rotating 3-dimensional view. **S4 Figure D1:** B. borealis, left IC tracts shown in blue, right IC tracts shown in red, minimum threshold set to 1% and maximum threshold set to 30% of waytotals. Orthographic view. **S4 Figure D2:** B. borealis, left IC tracts shown in blue, right IC tracts shown in red, set to a more liberal threshold of minimum 0.1% and maximum 5% of waytotals. Orthographic view. **S4 Figure D3:** B. borealis, left IC tracts shown in blue, right IC tracts shown in red, set to a more liberal threshold of minimum 0.1% and maximum 5% of waytotals. Still 3-dimensional view. **S4 Figure D4:** B. borealis, left IC tracts shown in blue, right IC tracts shown in red, set to a more liberal threshold of minimum 0.1% and maximum 5% of waytotals. Rotating 3-dimensional view. **S5 Figures. Contralateral collicular-cortical tractograms. S5 Figure A1:** D. delphis, left IC to right cortex tracts shown in green, right IC to left cortex tracts shown in pink, minimum threshold set to 1% and maximum threshold set to 30% of waytotals. Both cerebella excluded. Orthographic view. **S5 Figure A2:** D. delphis, left IC to right cortex tracts shown in green, right IC to left cortex tracts shown in pink, set to a more liberal threshold of minimum 0.1% and maximum 5% of waytotals. Both cerebella excluded. Orthographic view. **S5 Figure A3:** D. delphis, left IC to right cortex tracts shown in green, right IC to left cortex tracts shown in pink, set to a more liberal threshold of minimum 0.1% and maximum 5% of waytotals. Both cerebella excluded. Still 3-dimensional view. **S5 Figure A4:** D. delphis, left IC to right cortex tracts shown in green, right IC to left cortex tracts shown in pink, set to a more liberal threshold of minimum 0.1% and maximum 5% of waytotals. Both cerebella excluded. Rotating 3-dimensional view. **S5 Figure B1:** S. attenuata, left IC to right cortex tracts shown in green, right IC to left cortex tracts shown in pink, minimum threshold set to 1% and maximum threshold set to 30% of waytotals. Both cerebella excluded. Orthographic view. **S5 Figure B2:** S. attenuata, left IC to right cortex tracts shown in green, right IC to left cortex tracts shown in pink, set to a more liberal threshold of minimum 0.1% and maximum 5% of waytotals. Both cerebella excluded. Orthographic view. **S5 Figure B3:** S. attenuata, left IC to right cortex tracts shown in green, right IC to left cortex tracts shown in pink, set to a more liberal threshold of minimum 0.1% and maximum 5% of waytotals. Both cerebella excluded. Still 3-dimensional view. **S5 Figure B4:** S. attenuata, left IC to right cortex tracts shown in green, right IC to left cortex tracts shown in pink, set to a more liberal threshold of minimum 0.1% and maximum 5% of waytotals. Both cerebella excluded. Rotating 3-dimensional view. **S5 Figure C1:** L. acutus, left IC to right cortex tracts shown in green, right IC to left cortex tracts shown in pink, minimum threshold set to 1% and maximum threshold set to 30% of waytotals. Both cerebella excluded. Orthographic view. **S5 Figure C2:** L. acutus, left IC to right cortex tracts shown in green, right IC to left cortex tracts shown in pink, set to a more liberal threshold of minimum 0.1% and maximum 5% of waytotals. Both cerebella excluded. Orthographic view. **S5 Figure C3:** L. acutus, left IC to right cortex tracts shown in green, right IC to left cortex tracts shown in pink, set to a more liberal threshold of minimum 0.1% and maximum 5% of waytotals. Both cerebella excluded. Still 3-dimensional view. **S5 Figure C4:** L. acutus, left IC to right cortex tracts shown in green, right IC to left cortex tracts shown in pink, set to a more liberal threshold of minimum 0.1% and maximum 5% of waytotals. Both cerebella excluded. Rotating 3-dimensional view. **S5 Figure D1:** B. borealis, left IC to right cortex tracts shown in green, right IC to left cortex tracts shown in pink, minimum threshold set to 1% and maximum threshold set to 30% of waytotals. Both cerebella excluded. Orthographic view. **S5 Figure D2:** B. borealis, left IC to right cortex tracts shown in green, right IC to left cortex tracts shown in pink, set to a more liberal threshold of minimum 0.1% and maximum 5% of waytotals. Both cerebella excluded. Orthographic view. **S5 Figure D3:** B. borealis, left IC to right cortex tracts shown in green, right IC to left cortex tracts shown in pink, set to a more liberal threshold of minimum 0.1% and maximum 5% of waytotals. Both cerebella excluded. Still 3-dimensional view. **S5 Figure D4:** B. borealis, left IC to right cortex tracts shown in green, right IC to left cortex tracts shown in pink, set to a more liberal threshold of minimum 0.1% and maximum 5% of waytotals. Both cerebella excluded. Rotating 3-dimensional view. **S6 Figures. Contralateral collicular-cerebellar tractograms. S6 Figure A1.** D. delphis, left IC to right cerebellum tracts shown in orange, right IC to left cerebellum tracts shown in turquoise, minimum threshold set to 1% and maximum threshold set to 30% of waytotals. Orthographic view. **S6 Figure A2.** D. delphis, left IC to right cerebellum tracts shown in orange, right IC to left cerebellum tracts shown in turquoise, set to a more liberal threshold of minimum 0.1% and maximum 5% of waytotals. Orthographic view. **S6 Figure A3.** D. delphis, left

IC to right cerebellum tracts shown in orange, right IC to left cerebellum tracts shown in turquoise, set to a more liberal threshold of minimum 0.1% and maximum 5% of waytotals. Still 3-dimensional view. **S6 Figure A4.** D. delphis, left IC to right cerebellum tracts shown in orange, right IC to left cerebellum tracts shown in turquoise, set to a more liberal threshold of minimum 0.1% and maximum 5% of waytotals. Rotating 3-dimensional view. **S6 Figure B1.** S. attenuata, left IC to right cerebellum tracts shown in orange, right IC to left cerebellum tracts shown in turquoise, minimum threshold set to 1% and maximum threshold set to 30% of waytotals. Orthographic view. **S6 Figure B2.** S. attenuata, left IC to right cerebellum tracts shown in orange, right IC to left cerebellum tracts shown in turquoise, set to a more liberal threshold of minimum 0.1% and maximum 5% of waytotals. Orthographic view. **S6 Figure B3.** S. attenuata, left IC to right cerebellum tracts shown in orange, right IC to left cerebellum tracts shown in turquoise, set to a more liberal threshold of minimum 0.1% and maximum 5% of waytotals. Still 3-dimensional view. **S6 Figure B4.** S. attenuata, left IC to right cerebellum tracts shown in orange, right IC to left cerebellum tracts shown in turquoise, set to a more liberal threshold of minimum 0.1% and maximum 5% of waytotals. Rotating 3-dimensional view. **S6 Figure C1.** L. acutus, left IC to right cerebellum tracts shown in orange, right IC to left cerebellum tracts shown in turquoise, minimum threshold set to 1% and maximum threshold set to 30% of waytotals. Orthographic view. **S6 Figure C2.** L. acutus, left IC to right cerebellum tracts shown in orange, right IC to left cerebellum tracts shown in turquoise, set to a more liberal threshold of minimum 0.1% and maximum 5% of waytotals. Orthographic view. **S6 Figure C3.** L. acutus, left IC to right cerebellum tracts shown in orange, right IC to left cerebellum tracts shown in turquoise, set to a more liberal threshold of minimum 0.1% and maximum 5% of waytotals. Still 3-dimensional view. **S6 Figure C4.** L. acutus, left IC to right cerebellum tracts shown in orange, right IC to left cerebellum tracts shown in turquoise, set to a more liberal threshold of minimum 0.1% and maximum 5% of waytotals. Rotating 3-dimensional view. **S6 Figure D1.** B. borealis, left IC to right cerebellum tracts shown in orange, right IC to left cerebellum tracts shown in turquoise, minimum threshold set to 1% and maximum threshold set to 30% of waytotals. Orthographic view. **S6 Figure D2.** B. borealis, left IC to right cerebellum tracts shown in orange, right IC to left cerebellum tracts shown in turquoise, set to a more liberal threshold of minimum 0.1% and maximum 5% of waytotals. Orthographic view. **S6 Figure D3.** B. borealis, left IC to right cerebellum tracts shown in orange, right IC to left cerebellum tracts shown in turquoise, set to a more liberal threshold of minimum 0.1% and maximum 5% of waytotals. Still 3-dimensional view. **S6 Figure D4.** B. borealis, left IC to right cerebellum tracts shown in orange, right IC to left cerebellum tracts shown in turquoise, set to a more liberal threshold of minimum 0.1% and maximum 5% of waytotals. Rotating 3-dimensional view. **S7 Figures.** Specific cerebellar targets in IC-cerebellar tractograms. **S7 Figure A1.** D. delphis, left IC to right cerebellum tracts shown in orange, minimum threshold set to 1% and maximum threshold set to 30% of waytotals. **S7 Figure A2.** D. delphis, right IC to left cerebellum tracts shown in blue, minimum threshold set to 1% and maximum threshold set to 30% of waytotals. **S7 Figure B1.** S. attenuata, left IC to right cerebellum tracts shown in orange, minimum threshold set to 1% and maximum threshold set to 30% of waytotals. **S7 Figure B2.** S. attenuata, right IC to left cerebellum tracts shown in blue, minimum threshold set to 1% and maximum threshold set to 30% of waytotals. **S7 Figure C1.** L. acutus, left IC to right cerebellum tracts shown in orange, minimum threshold set to 1% and maximum threshold set to 30% of waytotals. **S7 Figure C2.** L. acutus, right IC to left cerebellum tracts shown in blue, minimum threshold set to 1% and maximum threshold set to 30% of waytotals. **S7 Figure D1.** B. borealis, left IC to right cerebellum tracts shown in orange, minimum threshold set to 1% and maximum threshold set to 30% of waytotals. **S7 Figure D2.** B. borealis, right IC to left cerebellum tracts shown in turquoise, minimum threshold set to 1% and maximum threshold set to 30% of waytotals. **S8 Tables.** Results of the systematic tractographical analysis conducted for each specimen. **S8 Table** 1. Common dolphin. **S8 Table 2.** Pantropical spotted dolphin. **S8 Table** 3. Atlantic white-sided dolphin. **S8 Table** 4. Sei whale. **S9 Table.** Cerebellar projection sites and putative functions. **S9 Table 1.** Display of which cerebellar subregions were targeted in auditory-cerebellar traces, with columns further specifying the species involved, laterality in odontocetes versus mysticete, and putative function as informed by studies in other non-cetacean mammals [35, 83-84]. Asterisks in the "Laterality in Odontoceti" column indicate that the listed laterality was not consistently observed across all specimens, but rather represents

the overall lateralization pattern. For example, the asterisk next to the "Left" designation in this column's "Vermis" row reflects that the evidence for a left-vermis bias in odontocete auditory-cerebellar traces was somewhat mixed, given its left-side targeting in S. attenuata and its bilateral targeting in L. acutus.
(ZIP)

## Acknowledgments

We gratefully acknowledge the International Fund for Animal Welfare (formerly the Cape Cod Stranding Network) for collecting the stranded *Lagenorhynchus acutus* that was scanned in this study. We also extend our thanks to Lori Marino, Ann Pabst, and William McLellan for the *Stenella attenuata* and *Delphinus delphis* specimens, and thank Karla Miller's colleagues at Oxford University for supporting our use of their SSFP imaging protocols. Finally, we thank the Republic of Iceland for supporting the collection and transport of the *Balaenoptera borealis* specimen.

## Author contributions

**Conceptualization:** Sophie Flem, Peter F. Cook.

**Data curation:** Sophie Flem, Gregory Berns, Ben Inglis, Dillon Niederhut, Eric Montie, Terrence Deacon, Karla L. Miller, Peter Tyack, Peter F. Cook.

**Formal analysis:** Sophie Flem, Karla L. Miller, Peter F. Cook.

**Funding acquisition:** Peter F. Cook.

**Investigation:** Sophie Flem, Terrence Deacon, Peter Tyack, Peter F. Cook.

**Methodology:** Sophie Flem, Gregory Berns, Ben Inglis, Dillon Niederhut, Eric Montie, Terrence Deacon, Karla L. Miller, Peter F. Cook.

**Project administration:** Sophie Flem, Terrence Deacon, Peter Tyack, Peter F. Cook.

**Resources:** Gregory Berns, Ben Inglis, Dillon Niederhut, Eric Montie, Terrence Deacon, Karla L. Miller, Peter Tyack, Peter F. Cook.

**Software:** Sophie Flem, Gregory Berns, Ben Inglis, Dillon Niederhut, Eric Montie, Terrence Deacon, Karla L. Miller, Peter F. Cook.

**Supervision:** Gregory Berns, Peter F. Cook.

**Validation:** Sophie Flem, Gregory Berns, Karla L. Miller, Peter F. Cook.

**Visualization:** Sophie Flem, Peter F. Cook.

**Writing – original draft:** Sophie Flem, Peter F. Cook.

**Writing – review & editing:** Sophie Flem, Gregory Berns, Ben Inglis, Dillon Niederhut, Eric Montie, Terrence Deacon, Karla L. Miller, Peter Tyack, Peter F. Cook.

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
