## [Decision Letter · Decision Letter 0]

PONE-D-24-37071Lateralized cerebellar connectivity differentiates auditory pathways in echolocating and non-echolocating whalesPLOS ONE

Dear Dr. Flem,

Thank you for submitting your manuscript to PLOS ONE. After careful consideration, we feel that it has merit but does not fully meet PLOS ONE’s publication criteria as it currently stands. Therefore, we invite you to submit a revised version of the manuscript that addresses the points raised during the review process.

The reviewer has identified several areas of the manuscript that require reworking and revision. Please address all comments in your revised work.

We look forward to receiving your revised manuscript.

Kind regards,

James J Cray Jr., Ph.D.

Academic Editor

PLOS ONE

2. We note that your Data Availability Statement is currently as follows: [All relevant data are within the manuscript and its Supporting Information files.] .Please confirm at this time whether or not your submission contains all raw data required to replicate the results of your study. Authors must share the “minimal data set” for their submission. PLOS defines the minimal data set to consist of the data required to replicate all study findings reported in the article, as well as related metadata and methods (https://journals.plos.org/plosone/s/data-availability#loc-minimal-data-set-definition).

Additional Editor Comments (if provided):

Reviewers' comments:

Reviewer's Responses to Questions

**Comments to the Author**

1. Is the manuscript technically sound, and do the data support the conclusions?

Reviewer #1: Partly

2. Has the statistical analysis been performed appropriately and rigorously? 

Reviewer #1: Yes

3. Have the authors made all data underlying the findings in their manuscript fully available?

Reviewer #1: Yes

4. Is the manuscript presented in an intelligible fashion and written in standard English?

Reviewer #1: No

5. Review Comments to the Author

Reviewer #1: General comment and overview:

As a preface, I would first like to say that it was difficult to make comments without line numbers so, please, in the next version it would be advisable to ask the authors to add line numbers to facilitate the reviewing process for everyone, thank you. I hope the authors can address the minor changes I will comment on.

This paper explores, thanks to ex-vivo brains scanned with SSFP DWI sequence, the descending and ascending acoustic pathways of three odontocetes and, for the first time, a mysticete. These kinds of studies in fact are essential nowadays since, for now, they are the only ones that could give us insights into the morphology and possibly the function of the brain (and related systems) of these species.

Although this paper provides partially already published data and a lot of new interesting data, it is very confusing, not only in terms of how the data were presented (and thus difficult to follow the text and the images) but also in terms of the basics of neuroanatomy.

The authors seem to confuse the motor pathways with the sensory/intracranial pathways and descending auditory pathways, sometimes overlapping them to discuss some possible function. One example of these misunderstandings between different systems is Figure 8. It appears that fibers run from canonical M1 in odontocetes (red arrows) and temporal auditory cortex in mysticete (blue arrow) to the ipsilateral IC to then decussate going to cerebellum and finally “targeting” either the nucleus facialis or the nucleus ambiguus. But the underlying logic is not clear. Are these descending auditory pathways? Are these motor pathways? Or are the authors presenting a novel pathway?

To explain my perplexities simply, in men and other animals:

1. The (descending) motor pathways include the pyramidal, extrapyramidal and other non-cortical motor tracts. Of these, the pathways involving the innervation of the nucleus ambiguus (thus reaching to the larynx in humans and why not possibly in whales) and facial motor nucleus (thus innervating the mimic muscles in humans and possibly monkey lips in dolphins) belong to the cortico-bulbar pathway which in turn is linked to the cortico-spinal tract (pyramidal).

2. On the other hand, the cerebellum generally plays a modulation role, integrating sensory feedback and motor coordination, thus (indirectly) to muscles involved in movement and posture maintenance, through connections to M1, to the thalamus and the rubrospinal tract. It certainly sends efferents but NOT directly to voluntary motor outputs or to nuclei related to voluntary activity, therefore NOT to nucleus ambiguus and/or facial nerve. It sends efferents (except for the ones to the midbrain and cortex) basically to vestibular nuclei through the inferior (or caudal) cerebellar peduncle.

3. The last point is about descending auditory pathways. These pathways do not not pass the brainstem and return via olivo-coclear fibers to the ear and they are mostly responsible for avoiding loud and annoying noises or sounds, at least for what is currently known in mammals.

All this information has been the results of acquisition of data over many years and by combining other techniques, including functional studies and there there are still doubts and uncovered points, especially when talking about the brainstem which is comparable to an intricate system where cells are in the middle of fibers of more or less variable size.

Unless the authors assume they discovered new pathways only thanks to DTI-tractography (which, by the way, compared to other DWI algorithms, is also much less performant in detecting crossing fibers), I do not personally think what they report is anatomically right. I do believe instead that those cerebellar-IC pathways might be most probably sensitive or part of an intracranial integrating loop which then goes to the motor or somatosensory cortex for the “final” echoes production, but even this needs to be explained and discussed.

I would therefore suggest the authors to divide the canonical pathways (found in reference human neuroanatomy textbooks) involved in ascending and descending hearing, integration, movement and so on to then overlap them to what they found on the dolphins and make a final image scheme which explains clearly and simply what are these results represent.

Other comments:

Abstract:

I would personally not start the abstract like this saying what the authors found. I can see Plos One suggests including the objectives as the first point but I think it is better to start with a small introduction and objectives following M&M, results + discussion.

Introduction:

- Page 5 lane 7: after the citation of “(Ladygina and Supin, 1977)” please consider also "Lende RA, Akdikmen S (1968) Motor field in cerebral cortex of the bottlenose dolphin. J Neurosurg 29:495–499. https://doi.org/10.3171/jns.1968.29.5.0495", "Lende RA, Welker WI (1972) An unusual sensory area in the cerebral neocortex of the bottlenose dolphin, Tursiops truncatus. Brain Res 45:555–560. https://doi.org/10.1016/0006-8993(72)90482-9" and "Sokolov VE, Ladygina TF, Supin AI (1972) Localization of sensory zones in the dolphin cerebral cortex. Doklady Akademii Nauk SSSR 202(2):490–493. http://www.ncbi.nlm.nih.gov/pubmed/4333815”. There are also retrograde tracing: “Garey LJ, Revishchin AV (1990) Structure and thalamocortical relations of the cetacean sensory cortex: histological, tracer and immunocytochemical studies. In: Thomas JA, Kastelein RA (eds) Sensory abilities of cetaceans. Springer, New York, pp 19–30. https://doi.org/10.1007/978-1-4899-0858-2_2.”

- Page 5 lane 7: across “mammalian” species.

- Page 5 lane 15: please after “subcortical auditory system” a citation is needed and I would suggest the studies of Montie et al 2007 and 2008. Montie, E. W., Schneider, G. E., Ketten, D. R., Marino, L., Touhey, K. E., & Hahn, M. E. (2007). Neuroanatomy of the subadult and fetal brain of the Atlantic white‐sided dolphin (Lagenorhynchus acutus) from in situ magnetic resonance images. The Anatomical Record: Advances in Integrative Anatomy and Evolutionary Biology: Advances in Integrative Anatomy and Evolutionary Biology, 290(12), 1459-1479. Montie, E. W., Schneider, G., Ketten, D. R., Marino, L., Touhey, K. E., & Hahn, M. E. (2008). Volumetric neuroimaging of the atlantic white‐sided dolphin (Lagenorhynchus acutus) brain from in situ magnetic resonance images. The Anatomical Record: Advances in Integrative Anatomy and Evolutionary Biology: Advances in Integrative Anatomy and Evolutionary Biology, 291(3), 263-282.

- Page 5 lane 17: the citation “Cozzi et al., 2018” is wrong throughout all the text. Since it is the atlas the one cited the correct citation is “Huggenberger et al., 2018”. (If the authors want to cite the book it is “Cozzi et al., 2017”)

- Page 7 lane 19: specify that Berns found that pathway because they seeded from the IC. Othewise when parcellated the thalamus they also found ascending auditory pathways to SS A1.

-Page 11 lane 5: I would delete Gerussi et al from the definition of “DTI tractography” as they used constrained spherical deconvolution. Or writing like “DTI and CSD tractography …”

Results:

- Page 12 lane 3: “made up” instead of “make” and following verb tenses from present to past tense.

- Page 14: is it possible to have Figure 1 and Table 2 joined? Like adding the table under the picture. This would group the arguments better without going back and forth. In figure 1 is “Auditory ascending” the IC - ipsilateral cortex?

- Page 22 Figure 3: It looks like figure 3b right the cerebrum is broken in the temporal side and also that pathways are going to suprasylvian A1, isn't it? Could the authors pleas also specify in all MRI pics if the images shown are from DWI or anatomical ones? It looks like a b are T1-anatomical while c and d are DWI, right?

Materials and methods:

- Page 53 from lane 5: “immediately prior”, how much before? Then please explain why 2% agarose and gadolinium. I know why but it is better for the reader to explain why.

- Page 53 lane 9: Please specify which carcass decomposition code.

- Page 53 lane 14: Since you did not write what organs one may think the pathologies affected the brain too. Therefore specify that it was not brain related or something like this.

- Page 53 paragraph from lane 16: It perfectly understandable that being in possession of the brain of such a rare species is important and needs to be studied. However, a more detailed description of the hunting regulations (if possible) and transportation of such brain is equally important. The use of a hunted animal species should also be overseen by an ethics committee, from a local research institute at least, even though it is not a CITES protected species.

6. PLOS authors have the option to publish the peer review history of their article (what does this mean?). If published, this will include your full peer review and any attached files.

Reviewer #1: No

---

## [Author Response · Author response to Decision Letter 1]

6 Mar 2025

Dear Dr. James J Cray Jr. and Reviewer One,

We greatly appreciate your effort in shepherding our manuscript through the submission and review process at PLOS One. We have read and closely considered all of Reviewer One’s comments, and address each in turn below. We would like to extend our thanks to our reviewer as well – this is a long and detailed paper, requiring substantial time commitment to work through, and the reviewer’s thoughtful points have substantially strengthened our paper.

First, we would like to address the reviewer’s primary concerns and how we revised our paper accordingly. Then we list smaller changes in order below.

Most importantly, the reviewer expressed clearly justified confusion about our terminology regarding the descending auditory pathways and potential efferent pathways:

“The authors seem to confuse the motor pathways with the sensory/intracranial pathways and descending auditory pathways, sometimes overlapping them to discuss some possible function. One example of these misunderstandings between different systems is Figure 8. It appears that fibers run from canonical M1 in odontocetes (red arrows) and temporal auditory cortex in mysticete (blue arrow) to the ipsilateral IC to then decussate going to cerebellum and finally “targeting” either the nucleus facialis or the nucleus ambiguus. But the underlying logic is not clear. Are these descending auditory pathways? Are these motor pathways? Or are the authors presenting a novel pathway?

…

Unless the authors assume they discovered new pathways only thanks to DTI-tractography (which, by the way, compared to other DWI algorithms, is also much less performant in detecting crossing fibers), I do not personally think what they report is anatomically right. I do believe instead that those cerebellar-IC pathways might be most probably sensitive or part of an intracranial integrating loop which then goes to the motor or somatosensory cortex for the “final” echoes production, but even this needs to be explained and discussed.”

We fully understand the reviewer’s points here and want to clarify up front in this letter: we do not claim to have discovered new pathways from our tractography. Nor did we specifically trace any efferent pathways. To clarify what we did trace, and to provide a visual reference point or “key” to help readers keep track of the details of the traces described in the results and discussion sections, we have added a new figure at the start of the results section. Figure 1a specifies the nomenclature, ROIs, and inclusion/exclusion criteria used for each tracing protocol. In our initial submission, we did not clearly define our terminology up front, and used the terms “descending acousticomotor” and “descending auditory” interchangeably. Because we make a number of inferences about sound production in dolphins and whales, we frequently discuss the integration of acoustic input and motor output, and did not always clearly specify what was inferred vs what was measured. In the current study we traced only from IC to cortex and IC to cerebellum. We assume these are auditory pathways. Further, as in Huffman (1990) we assume the IC-to-cerebellum pathway transmits descending acoustic information. Certainly, in cerebellum, we further assume auditory pathways converge with motor pathways for sensorimotor integration, which we strongly suggest bears on echolocation in odontocetes. We do note, incidentally, that our IC to cortex and IC to cerebellum pathways sometimes terminate in putative motor cortex in the odontocetes, and that in mysticete and odontocetes our IC-cerebellum pathways sometimes terminate in extra-cerebellar brainstem locations that may match putative vocal-motor nuclei. But we did not segment putative auditory or motor cortex, nor brainstem nuclei, and so did not set these as specific tracing targets. We now clarify this in the introduction (particularly lines 183-193 and 219-236), in the newly-created Figure 1a, in Figure 8 and its caption, and in the discussion (particularly lines 590-598 and 754-757). We further specify that we are not tracing or directly addressing the well-established olivocochlear descending auditory pathways that may deal with response to loud/annoying sounds the reviewer brings up. The putative “descending” auditory pathway we are examining is between IC and the cerebellum.

Further confusion likely stemmed from the initial organization of Figure 8, showing putative pathways from cerebellum to brainstem phonatory nuclei. We recognize that cerebellar output to volitional motor systems is typically understood to travel to M1 via the thalamus, and now include this putative (but not traced in our study) pathway in Figure 8. However, the exact pathways by which the cerebellum influences motor output in different species are still being studied. While there is clear evidence in mammals of cerebellar output via thalamus directly to motor cortex - and from there via the pyramids to upper motor nuclei, and via rubrospinal tract to the spine – newer evidence also indicates that the cerebellum targets essentially all upper motor neurons, including in brainstem nuclei (Novello et al., 2024). There is explicit experimental evidence from gold standard chemical tracing studies that the cerebellum directly targets midbrain nuclei involved in phonation, such as the periaqueductal grey, and brainstem nuclei involved in facial and throat motor control such as the solitary tract nucleus (Asanuma et al., 1983; Judd et al., 2021). There is some evidence that it may target ambiguus directly (Moolenaar & Rucker, 1976), but this is based on sparse data and is still debated.

Therefore, we have updated Figure 8 to include pathways from the cerebellum back to motor cortex via the thalamus (not explicitly traced), intracortical integrative pathways between A1 and M1 (not explicitly traced but potentially incidentally described), and M1 to phonatory nuclei (not explicitly traced), and also still include putative pathways from cerebellum either directly or indirectly to phonatory brainstem nuclei (not explicitly traced but potentially incidentally described (see lines 511-519)). We think this latter inclusion is important both because A. these are future targets for tractography examining cetacean vocal behavior, and B. because the potential for direct synaptic paths from the cerebellum to phonatory nuclei in the brainstem might bear on the temporal resolution of echolocation and related sensorimotor integration and predictive coding. We further emphasize that this is a model figure, not a direct representation of our tract tracing.

The reviewer’s comments were helpful in clarifying our findings and terminology. Importantly, although we did not explicitly trace efferent pathways, nor do we claim to have, we continue to emphasize the likely relevance of the IC-to-cerebellum pathway to vocal motor output. The cerebellum plays an important role in vocal signalling in birds and mammals (Pidoux et al., 2018; Smaers et al., 2018), and is even increasingly shown to be involved in human language processing (Murdoch, 2010; Nakatani et al., 2022; Parker Jones et al., 2024). While the old view of the cerebellum as involved in postural accommodation and balance is not wrong, it has been greatly updated in recent years to emphasize the further importance of the cerebellum for prediction across behavioral and cognitive domains (Baumann et al., 2014; Ebner & Pasalar, 2008; Hull, 2020; Ishikawa et al., 2016; Klein et al., 2016). Our suggestion, emphasized in the introduction and conclusion, that the IC to cerebellum pathway is highly relevant to feed forward control and modulation of vocal behavior, both echolocation and communication, draws on a wealth of evidence that this descending pathway is highly relevant to vocal output across species. Much of this work is summarized in Huffman & Henson (1990), which we newly emphasize in the introduction in lines 219-236.

We have further emphasized the likely validity of our IC to cerebellar tracings in the conclusion – although the reviewer is certainly correct that DWI is not optimally suited for resolving crossing fibers, our data are high resolution, high angle (52 directions), have excellent signal to noise, and implement a two diffusion direction crossing fiber model that has shown good success at resolving complex crossing fibers (Jbabdi et al., 2010).

Other, more minor edits and revisions are as follows:

Preface- no line numbers

Response: Apologies, I definitely understand the difficulty of revising without line numbers. In the actual manuscript outside of the PLOS system, there are page numbers, but for some reason they didn’t show up in the final PDF file produced by the PLOS system. I’ll try to figure out how to preserve the line numbers next time, as I agree they are an important component!

Abstract

“I would personally not start the abstract like this saying what the authors found. I can see Plos One suggests including the objectives as the first point but I think it is better to start with a small introduction and objectives following M&M, results + discussion.”

Response: We appreciate the suggestion here, and made sure that each subsection (ex. M&M, results, etc.) was represented at least to some degree in our abstract, but are opting to retain the beginning structure suggested by PLOS One. However, because you brought our attention to improving the abstract, we did work to remove excess detail and simplify its contents.

Introduction

Add references to Page 5, Line 7 parenthetical citation

Reponse: Excellent suggestions! These have been added.

“- Page 5 lane 7: across “mammalian” species.”

Response: Added.

Page 5 line 15, add references after “subcortical auditory systems”

Response: Good catch– I took your suggestion on Montie et al. (2007) because this study reports on subcortical auditory structures. I decided to hold off on adding the Montie et al. (2008) paper here because it doesn’t meaningfully report on subcortical auditory structures; however, because it includes measurements of cerebellar volume, it is cited elsewhere in our manuscript. Instead, I opted to add two other foundational MRI studies from the 2000’s that do mention hypertrophy of subcortical auditory structures: Marino et al. (2002) and Oelschläger et al. (2007).

Page 5 line 17 and throughout, correct the Cozzi et al., 2018 citation

Response: Thanks for the correction. I changed all parenthetical citations to (Cozzi et al., 2017), and changed the bibliographic citation to reflect that we only used Chapter 6 of the Anatomy book– I did not actually use the Atlas companion up to this point! So thank you for the suggestion / clarification there, too.

“Page 7 lane 19: specify that Berns found that pathway because they seeded from the IC. Othewise when parcellated the thalamus they also found ascending auditory pathways to SS A1”

Response: Per your advice, I did add a sentence to this section specifying the findings of Berns et al. (2015) a bit more. But I did not add exactly what you requested, because in Berns et al. (2015) they only observed those robust suprasylvian projections when they seeded the temporal lobe and set suprasylvian A1 as a waypoint. The thalamic parcellation portion of the study actually found that when you probe the thalamic connections of deep temporal lobe A1, suprasylvian “A1,” and V1, only deep temporal A1 targeted MGN/ventrocaudal thalamus, while the suprasylvian “A1” and V1 paths through thalamus overlapped substantially in a more dorsal area.

“Page 11 lane 5: I would delete Gerussi et al from the definition of “DTI tractography” as they used constrained spherical deconvolution. Or writing like “DTI and CSD tractography …”

Response: Good distinction! Given the “flow” of the introduction, there weren’t many places to add and explain CSD tractography, so I simply removed the reference here.

Results

“Page 12 lane 3: “made up” instead of “make” and following verb tenses from present to past tense.”

Response: I changed the verb tenses in the places you mentioned, and double checked the rest of the results section to verify all verbs were in the past tense. Thanks!

“Page 14: is it possible to have Figure 1 and Table 2 joined? Like adding the table under the picture. This would group the arguments better without going back and forth. In figure 1 is “Auditory ascending” the IC - ipsilateral cortex?”

Response: This one is a bit complicated and is now more clearly laid out in the new Figure 1a. “Ascending Auditory” is slightly broader than “IC - Ipsilateral Cortex”; the tracing protocol for the former involved simply seeding the IC, while the latter involved seeding the IC and excluding any tracts that contacted the cerebellum or crossed the midline to the other hemisphere. The quantitative results of both traces are in supplemental table S8. However, the quantitative results of the simple IC seed trace are non-informative, as they just represent the total number of streamlines generated by Probtrackx (5000*number of voxels in seed region) corrected by total brain volume. Any single seed tracing in Probtrackx will always generate a waytotal equal to the number of probabilistic streamlines generated by voxel times number of voxels. In multi-region tracings, as in the IC with the cerebellum/interhemispheric exclusion, the waytotal IS informative, as it is the percentage of streamlines that meet the criteria of not entering cerebellum or crossing midline. Nonetheless, you are absolutely right to notice the discrepancy; we discussed and carefully considered how to create informative qualitative traces and quantitative traces when planning and executing data analysis for this paper. In light of this and other concerns you brought to our attention, we added an extra figure (Figure 1a) at the beginning of the results section to more explicitly define the naming system and ROI inclusion/exclusion criteria for each trace. Your question here also prompted us to find and fix any other inconsistencies in naming conventions across figures– so thank you again for pointing this out!

“Page 22 Figure 3: It looks like figure 3b right the cerebrum is broken in the temporal side and also that pathways are going to suprasylvian A1, isn't it? Could the authors pleas also specify in all MRI pics if the images shown are from DWI or anatomical ones? It looks like a b are T1-anatomical while c and d are DWI, right?”

Response to first question: Indeed, the sei whale brain had a section of temporal lobe removed for histology; since the removal didn’t penetrate too deep into major white matter tracts, and because of the rarity of such a well-preserved baleen brain, we decided to accept this limitation. However, you rightfully noticed that we failed to include this in the Materials and Methods section, so I went ahead and added this information there!

On suprasylvian A1 comment: Not really– at the most stringent thresholds, only the temporal lobe immediately lateral to MGN is reached, and in slightly lower thresholds like this one, projections run caudal but not necessarily dorsal, and thus remain in likely temporal lobe, not reaching a dorsal-enough location to qualify as suprasylvian, especially in the sei whale (though some of the odontocetes do display such a projection, especially at lower thresholds). Perhaps due to the cross-section angle it is hard to see this, but you can review the 3D images and GIFs of the brains to get a better visual on this.

Response to second question: Yes, a and b are high-resolution T1-weighted images while c and d are low-resolution b0 images. Per your suggestion, I added this information to the caption for the figure.

Materials and methods

“Page 53 from lane 5: “immediately prior”, how much before? Then please explain why 2% agarose and gadolinium. I know why but it is better for the reader to explain why.”

Response to first question: Per your advice, I changed “immediately” to “thirty minutes” in order to provide more specificity.

Response to second comment: Per your advice, I added a brief explanation and references at the end of that sentence.

“Page 53 lane 9: Please specify which carcass decomposition code.”

Response: Again, I was not

---

## [Decision Letter · Decision Letter 1]

Lateralized cerebellar connectivity differentiates auditory pathways in echolocating and non-echolocating whales

PONE-D-24-37071R1

Dear Dr. Flem,

We’re pleased to inform you that your manuscript has been judged scientifically suitable for publication and will be formally accepted for publication once it meets all outstanding technical requirements.

Kind regards,

James J Cray Jr., Ph.D.

Academic Editor

PLOS ONE

Additional Editor Comments (optional):

Reviewers' comments:

Reviewer's Responses to Questions

**Comments to the Author**

1. If the authors have adequately addressed your comments raised in a previous round of review and you feel that this manuscript is now acceptable for publication, you may indicate that here to bypass the “Comments to the Author” section, enter your conflict of interest statement in the “Confidential to Editor” section, and submit your "Accept" recommendation.

Reviewer #1: All comments have been addressed

2. Is the manuscript technically sound, and do the data support the conclusions?

Reviewer #1: Yes

3. Has the statistical analysis been performed appropriately and rigorously? 

Reviewer #1: Yes

4. Have the authors made all data underlying the findings in their manuscript fully available?

Reviewer #1: Yes

5. Is the manuscript presented in an intelligible fashion and written in standard English?

Reviewer #1: Yes

6. Review Comments to the Author

Reviewer #1: I thank the Authors for their great effort in improving the manuscript, making it more complete and correct.

7. PLOS authors have the option to publish the peer review history of their article (what does this mean?). If published, this will include your full peer review and any attached files.

Reviewer #1: No

---

## [Editor Report · Acceptance letter]

PONE-D-24-37071R1

PLOS ONE

Dear Dr. Flem,

I'm pleased to inform you that your manuscript has been deemed suitable for publication in PLOS ONE. Congratulations! Your manuscript is now being handed over to our production team.

Kind regards,

on behalf of

Dr. James J Cray Jr.

Academic Editor

PLOS ONE